There are amendments to this paper

# scRNA-seq in medulloblastoma shows cellular heterogeneity and lineage expansion support resistance to SHH inhibitor therapy

Jennifer Karin Ocasio [1,2,17], Benjamin Babcock [1,3,17], Daniel Malawsky[1], Seth J. Weir[1], Lipin Loo[2,4], Jeremy M. Simon [2,4,5], Mark J. Zylka[2,4,5], Duhyeong Hwang[6], Taylor Dismuke[1], Marina Sokolsky[6], Elias P. Rosen [6], Rajeev Vibhakar[7,8], Jiao Zhang[9,10], Olivier Saulnier [9,10], Maria Vladoiu[9,10], Ibrahim El-Hamamy[11,12], Lincoln D. Stein[11,12], Michael D. Taylor [9,10,13], Kyle S. Smith[14], Paul A. Northcott [14], Alejandro Colaneri[1,3], Kirk Wilhelmsen [1,3,15,18]* & Timothy R. Gershon [1,2,5,16,18]*

Targeting oncogenic pathways holds promise for brain tumor treatment, but inhibition of Sonic Hedgehog (SHH) signaling has failed in SHH-driven medulloblastoma. Cellular diversity within tumors and reduced lineage commitment can undermine targeted therapy by increasing the probability of treatment-resistant populations. Using single-cell RNA-seq and lineage tracing, we analyzed cellular diversity in medulloblastomas in transgenic, medulloblastoma-prone mice, and responses to the SHH-pathway inhibitor vismodegib. In untreated tumors, we find expected stromal cells and tumor-derived cells showing either a spectrum of neural progenitor-differentiation states or glial and stem cell markers. Vismodegib reduces the proliferative population and increases differentiation. However, specific cell types in vismodegib-treated tumors remain proliferative, showing either persistent SHH-pathway activation or stem cell characteristics. Our data show that even in tumors with a single pathway-activating mutation, diverse mechanisms drive tumor growth. This diversity confers early resistance to targeted inhibitor therapy, demonstrating the need to target multiple pathways simultaneously.

---

[1] Department of Neurology, University of North Carolina School of Medicine, Chapel Hill, NC 27599, USA. [2] UNC Neuroscience Center, University of North Carolina School of Medicine, Chapel Hill, NC 27599, USA. [3] Department of Genetics, University of North Carolina School of Medicine, Chapel Hill, NC 27599, USA. [4] Department of Cell Biology and Physiology, University of North Carolina School of Medicine, Chapel Hill, NC 27599, USA. [5] Carolina Institute for Developmental Disabilities, University of North Carolina School of Medicine, Chapel Hill, NC 27599, USA. [6] UNC Eshelman School of Pharmacy, University of North Carolina School of Medicine, Chapel Hill, NC 27599, USA. [7] Department of Pediatrics, University of Colorado Anschutz Medical Campus, Aurora, CO, USA. [8] Morgan Adams Foundation Pediatric Brain Tumor Research Program, Children's Hospital Colorado, Aurora, CO, USA. [9] Developmental & Stem Cell Biology Program, The Hospital for Sick Children, Toronto, ON M5G 0A4, Canada. [10] The Arthur and Sonia Labatt Brain Tumour Research Centre, The Hospital for Sick Children, Toronto, Ontario M5G 0A4, Canada. [11] Department of Molecular Genetics, University of Toronto, Toronto, ON M5G 0A4, Canada. [12] Program in Computational Biology, Ontario Institute for Cancer Research, Toronto, ON M5G 0A3, Canada. [13] Division of Neurosurgery, The Hospital for Sick Children, Toronto, ON M5S 3E1, Canada. [14] Department of Developmental Neurobiology, St. Jude Children's Research Hospital, Memphis, TN, USA. [15] Renaissance Computing Institute at UNC (RENCI), Chapel Hill, NC 27517, USA. [16] Lineberger Comprehensive Cancer Center, University of North Carolina School of Medicine, Chapel Hill, NC 27599, USA. [17]These authors contributed equally: Jennifer Karin Ocasio, Benjamin Babcock. [18]These authors jointly supervised this work: Kirk Wilhelmsen, Timothy R. Gershon. *email: kirk@med.unc.edu; gershont@neurology.unc.edu

Therapies that target tumor-initiating oncogenes have been highly effective for a small number of cancers including chronic myelogenous leukemia[1–3] and basal cell carcinoma[4], but have not been as broadly effective as hoped. Cellular diversity within tumors may critically limit the efficacy of targeted therapies. Cell-to-cell variability within tumors is readily studied using single-cell transcriptomic analysis[5–10], and prior studies suggest ways that diversity may contribute to treatment failure. For example, single-cell analysis of glioblastomas has shown that tumors contain subpopulations with diverse receptor tyrosine kinase (RTK) mutations, suggesting that no single RTK inhibitor would be likely to inhibit all tumor cells[11]. High-throughput single-cell transcriptomic analysis of Sonic Hedgehog (SHH)-driven medulloblastomas before and after initiation of SHH inhibitor may determine whether intra-tumor heterogeneity contributes to the process of therapeutic failure.

Medulloblastoma is among the most frequent malignant brain tumors in children[12]. Thirty percent of medulloblastomas show hyperactivation of the SHH signaling pathway[13,14], which also drives the normal proliferation of cerebellar granule neuron progenitors (CGNPs) during cerebellar development[15–17]. Activating SHH-pathway mutations in CGNPs cause medulloblastomas in genetically engineered mice, identifying CGNPs as the cell of origin for SHH-driven medulloblastoma and providing a primary animal model of the human disease[18–21].

Small-molecule inhibitors of SHH signaling have been investigated as potential therapies for SHH-driven cancers[22]. The Smoothened (SMO) inhibitor vismodegib, which disrupts a membrane-bound component of the SHH, has been shown to be effective for basal cell carcinoma and is FDA-approved for this purpose[23]. However, while vismodegib can induce initial responses in SHH-driven medulloblastoma, the long-term efficacy of vismodegib has been limited by the emergence of resistance during therapy[24–26]. Here, we use medulloblastomas that form in transgenic, *Smo*-mutant mice to study the early effects of vismodegib on cellular diversity in SHH-driven tumors and to determine if this diversity may contribute treatment failure.

## Results

**Vismodegib transiently slows medulloblastoma growth in mice.** We generated medulloblastoma-bearing mice by breeding the transgenic *SmoM2* mouse line, which harbors a mutant, constitutively active allele of *Smo*, preceded by a *LoxP-STOP-LoxP* sequence[27] with *Math1-Cre* mice, that express Cre recombinase in CGNPs, driven by the *Atoh1* (aka *Math1*) promoter[28–30]. The resulting *Math1-Cre/SmoM2* (*M-Smo*) mice developed medulloblastoma with 100% frequency by postnatal day 12 (P12). We administered either vismodegib or vehicle to medulloblastoma-bearing P12 *M-Smo* mice, daily from P12 to P15, and then every other day until symptomatic progression. Initially, vismodegib induced transient tumor regression, with reduced expression of phosphorylated RB (pRB; Fig. 1a, b). However, by 2 weeks on treatment, the fraction of pRB+ cells stopped declining and began to rise (Fig. 1b), and prolonged treatment did not significantly increase *M-Smo* mouse survival (Fig. 1c). For longitudinal measurement of pharmacodynamic response, we administered vismodegib to another medulloblastoma-prone genotype, *hGFAP-Cre/SmoM2/Gli-luc*, that carries a synthetic, SHH-sensitive luciferase reporter construct (*G-Smo/Gli-luc*; Fig. 1d). Luciferase imaging showed that the first dose of vismodegib decreased SHH activation, but that

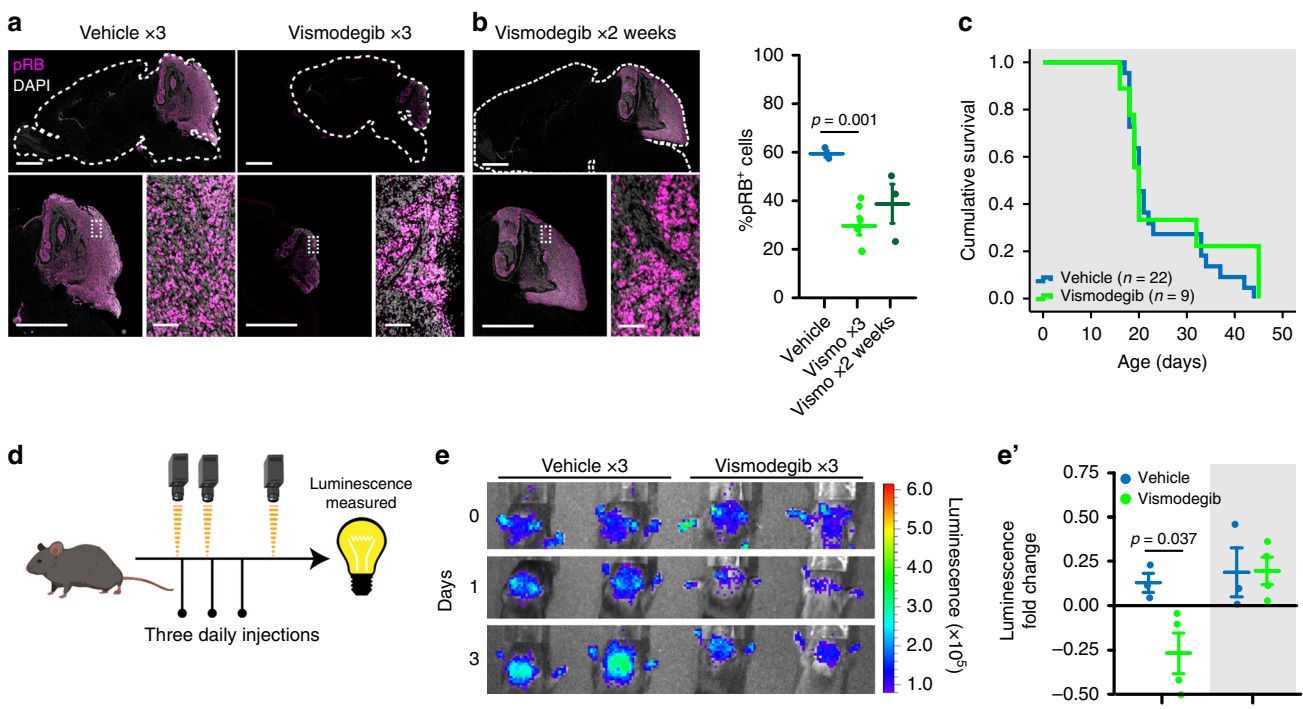

**Fig. 1 Vismodegib induces initial tumor response followed by rapid recurrence. a**, **b** Medulloblastoma in sagittal hindbrain sections stained for pRB, from representative **a** P15 *M-Smo* mice, treated with three daily doses of vehicle or vismodegib, or **b** P25 M-Smo mice treated for 2 weeks with vismodegib, and quantification of pRB+ fraction in the indicated groups. **c** Kaplan–Meier curve comparing survival of *M-Smo* mice treated with vismodegib or vehicle. **d** Schematic showing the timing of luciferase imaging and vismodegib administration. **e** Luciferase signal driven by *Gli-luc* at indicated times. **e′** Luminescence fold change over the indicated intervals in replicate vismodegib-treated and control mice. Each dot represents the value for a specific replicate animal. Horizontal lines indicate the means, and error bars indicate SEM. *p* Values determined by two-sided Student's *t*-test. Scale bars = 2 mm, except in insets, where scale bars = 50 μm.

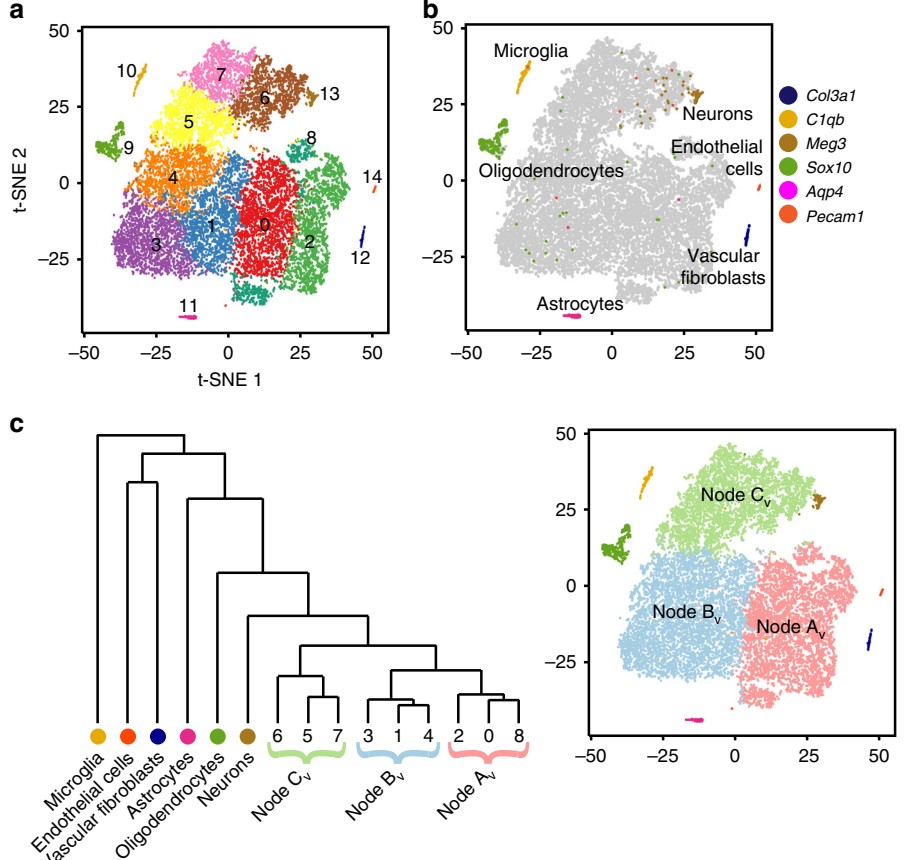

**Fig. 2 Single-cell transcriptomic analysis identifies discrete types of cells. a** t-SNE projection of cells from five vehicle-treated tumors, with cells placed according to Euclidean distance from each other in ten-dimensional PCA. The color code indicates clusters. **b** Feature plot of indicated vascular, microglial and glial markers on the t-SNE projection in **a**. **c** Dendrogram showing relationship of Nodes $A_V$–$C_V$ defined by HCA and t-SNE projection as in **a**, color-coded for Nodes and identified cell types. Thresholds for all feature plots are listed in Supplementary Dataset 9.

SHH activity progressively increased by the third day of treatment, (Fig. 1e, e′). Prior studies have associated vismodegib failure with tumor stem cells, defined by SOX2 expression, lineage tracing and transplantation experiments[31]. To gain further information on how cellular diversity contributes to resistance, we subjected tumors from *M-Smo* mice high-throughput, single-cell transcriptomic analysis, and compared tumors in the early stages of vismodegib therapy to vehicle-treated controls.

**Drop-seq analysis identifies stromal and tumor cells**. We treated two groups of five P12 *M-Smo* mice with three daily doses of either vismodegib or vehicle, then harvested tumors from all ten mice at P15. Tumors were dissociated and using the Drop-seq protocol V3.1 (ref. [32]), individual cells co-encapsulated in a microfluidics chamber with primer-coated beads, allowing mRNAs to be tagged with cell-specific bar codes and then amplified for library construction. After sequencing, transcript identities were determined by the 3′ UTR sequence and matched to cell identities determined from the bead-specific barcodes. We considered each bead-specific barcode to represent a putative cell, and we analyze all putative cells that met inclusion criteria described in Supplementary Materials and Methods, to address the common problems of gene drop out, unintentional cell–cell multiplexing and premature cell lysis[33,34]. A total of 84% of putative cells met inclusion criteria and were included as informative cells in the analysis.

To assess baseline cellular diversity, we analyzed the cells collected from vehicle-treated tumors. We conducted a principal component analysis (PCA) of the ~1500 genes that showed the highest cell–cell variation, defined by the magnitude of mean expression and dispersion (variance/mean). The first 11 principal components (PCs) were selected for further analysis. We rejected one PC, PC10, that highlighted batch effect variables, consistent with prior published approaches, proceeding with analysis on 10 PCs[35]. Louvain clustering on the PC-derived Shared Nearest Neighbor graph divided the cells into 15 clusters. Concurrently with cluster analysis, we applied t-distributed Stochastic Neighbor Embedding (t-SNE) to the PCs to project each cell into a two-dimensional graph according to local similarities, and lastly, color-coded cells according to their cluster identities (Fig. 2a). To analyze the biological relevance of the clusters, we generated cluster-specific differential expression profiles (Supplementary Dataset 1) by comparing the expression of each gene by cells within each cluster versus all cells outside the cluster. We then searched these profiles for specific markers to infer the biological identity of cells within each cluster.

These methods defined six clusters that localized to discrete regions of the t-SNE projection, without closely apposed neighbors. The gene expression patterns of these clusters identified them as endothelial cells, vascular fibroblasts, microglia, neurons, oligodendrocytes and astrocytes (Fig. 2b, Supplementary Dataset 1). Each of these cell types is expected in the stroma within or adjacent to the tumors. The other nine clusters were

closely apposed in a cohesive, multi-cluster complex, suggesting a spectrum of cells in transition between different states.

To develop an ordered classification within the multi-cluster complex, we applied hierarchal cluster analysis (HCA) to the centroids of the PCA-defined clusters. The resulting dendrogram separated the six clusters of cells with stromal characteristics from the rest of the population (Fig. 2c). The other nine clusters subdivided into two subsets, one of which split again before individual clusters emerged, highlighting three key branch points (Fig. 2c). We designated these branch points as Nodes $A_{Vehicle}$, $B_{Vehicle}$ and $C_{Vehicle}$ ($A_V$–$C_V$). We next assessed the resemblance of the cells in each cluster and node to the cells of the normal cerebellum during the period of postnatal neurogenesis.

**Differentiation spectrum approximates Atoh1 lineage.** In order to compare medulloblastomas to the normal tissue of origin, we used Drop-seq to generate single-cell transcriptomic data from the early postnatal cerebellum. We studied cerebella from five wild-type (WT) mice at P7, a time with an abundance of both proliferating and differentiating CGNPs. Single-cell analysis performed as in the M-Smo analysis divided 7090 WT cells that passed QC (70% of putative cells) into 14 clusters (Supplementary Fig. 1a). We generated cluster-specific differential expression profiles (Supplementary Dataset 2) and identified cell types with recognizable patterns of gene expression (Fig. 3a). Six clusters defined discrete stromal cell types, including endothelial cells, vascular fibroblasts, microglia, Purkinje neurons, oligodendrocytes and astrocytes.

Two subsets of cells formed separate multi-cluster groupings, identified as neural progenitors in a spectrum of differentiation states by complementary patterns of expression of the proliferation marker *Mki67* and the neuronal differentiation marker *Meg3*. One multi-cluster population expressed *Pax3* and *Pax2*, identifying it as GABAergic interneuron progenitors and neurons, as described in previous work[36]. In this GABAergic group, we noted sequential expression of the transcription factors *Ascl1*, *Sox2*, *Pax3* and *Pax2*, paralleling the progression from proliferation to differentiation (Fig. 3b). The other multi-cluster population showed gene expression patterns identifying it as predominantly CGNPs in a range of differentiation states, from proliferative, undifferentiated cells expressing the SHH-pathway transcription factor *Gli1*, to cells in successive states of CGN differentiation marked by sequential expression of markers *Ccnd2*, *Barhl1*, *Cntn2*, *Rbfox3* and *Grin2b* (Fig. 3c). Differentiated neurons were subclassified in the WT and *M-Smo* datasets using glutamatergic marker *Slc17a6* (aka *vGlut2*), GABAergic marker *Gad1* (aka *Gad67*), Purkinje neuron marker *Calb1*, the CGN marker *Calb2*, and *Eomes* (aka *Tbr2*), which distinguished the small population of unipolar brush cells (UBCs) from the molecularly similar CGNs (Supplementary Fig. 1b, c).

We analyzed the similarities between cells from *M-Smo* tumors and WT P7 cerebella. The Seurat canonical correlation analysis (CCA) provides an implementation for integrating single-cell data across different datasets by projecting the datasets into a maximally correlated lower dimensional subspace based on common sources of variation[37]. Using Seurat CCA, we generated a t-SNE projection that included both P7 WT and P15 *M-Smo* cells (Fig. 3d). In this t-SNE, stroma cell types from tumors and from WT cerebella co-clustered together, while progenitor-like tumor cells of Nodes $A_V$–$C_V$ clustered separately from all WT clusters, forming a multi-cluster group alongside the CGNPs, with a parallel axis of differentiation, and with the GABAergic progenitors and neurons arrayed in another group on the other side of the CGNPs. From this analysis, we conclude that the transcriptomes of the stromal cells types from tumors and WT

were highly similar, while the cells of Nodes $A_V$–$C_V$ were significantly dissimilar to all cell types of the WT cerebella.

In order to look beyond the differences between progenitor-like tumor cells and WT progenitors, and instead focus on similarities, we used the *k*-nearest neighbor (k-NN) algorithm to project the P15 *M-Smo* cells into the WT P7 t-SNE (Fig. 3e). This algorithm projected *M-Smo* cells into the WT t-SNE according to their similarity to the cell types present in the WT dataset and positioned each *M-Smo* cell in the centroid of its three most similar WT cells. Like Seurat CCA, this method correctly matched cells from each *M-Smo* stromal cluster to the same cell type in the WT cerebellum, validating the approach (Fig. 3d, e). The cells of Nodes $A_V$–$C_V$ predominantly localized to the multi-cluster CGNP group, and mapped in an orderly progression to successively more differentiated regions (Fig. 3e). Of the cell types present in the P7 WT cerebellum, the cells of Nodes $A_V$–$C_V$ most closely matched the CGNPs and showed a range of differentiation states that parallel the CGNP developmental trajectory.

We extended the kNN comparison to WT cells at a range of ages, using published single-cell RNA-seq analyses of WT cells from cerebella and hindbrain at ages ranging from embryonic day 10 (E10) to P10[38,39]. We subjected the gene expression data from each of these studies to PCA analysis and t-SNE projection using our pipeline. In the dataset from Vladoiu et al.[34], markers identified the stromal cell types that we identified in our P7 WT studies (Supplementary Fig. 2a). In the dataset from Carter et al.[38,39], all stromal cell types other than neurons and astrocytes had been excluded prior to our analysis; as expected, our markers identified only the neurons and astrocytes in the corresponding WT t-SNE (Supplementary Fig. 2b). In both cell types, we identified a domain consisting of the *Atoh1* lineage, including CGNPs, CGNs and UBCs through a combination of markers *Atoh1*, *Barhl1*, *Grin2b* and *Eomes*; this domain predominantly comprised cells from E14–P10 mice (Supplementary Fig. 2a, b). We then used the kNN method to project either P7 WT cells from our study, or P15 *M-Smo* cells into the t-SNEs.

The P7 WT CGNPs projected into the regions of the embryonic–postnatal WT t-SNEs that corresponded with the *Atoh1* lineage, marked by expression of *Atoh1*, *Barhl1* and *Grin2b* (Fig. 3f, h). Similarly, Nodes $A_V$–$C_V$ *M-Smo* cells projected predominantly to this same region, in a progression that followed the trajectory of CGNP differentiation (Fig. 3g, i). However, some cells from Nodes $A_V$ and $B_V$ mapped to cells from earlier time points outside the domain of the Atoh1 lineage suggesting a more primitive state (Fig. 3g, i). Taken together, the Seurat CCA and kNN comparisons to WT cerebella show that *M-Smo* tumors contained progenitor-like cells that were clearly different from normal progenitors but mirrored a spectrum of differentiation states ranging from primitive neuroepithelial cells through the CGNPs with progressively greater differentiation.

**Two discrete fate trajectories within *M-Smo* tumors.** To further explore latent factors within the transitional states within *M-Smo* tumors, we used independent component analysis (ICA). For this analysis, we selected the population of medulloblastoma cells defined by Nodes $A_V$–$C_V$ and selected the genes that met variability criteria in this specific sub-population. We found that generating four ICs (IC $1_V$−$4_V$) produced robust components that identified non-overlapping sets of cells at each component's extremes (Supplementary Dataset 3). One of these ICs, IC $2_V$ was defined by differentiation state, with neural differentiation markers highly represented in the 50 most heavily weighted genes (Supplementary Dataset 3). Using all four ICs as dimensions, we mapped the medulloblastoma cells in a new, IC-based t-SNE projection (Supplementary Fig. 3a). In this projection, the Nodes

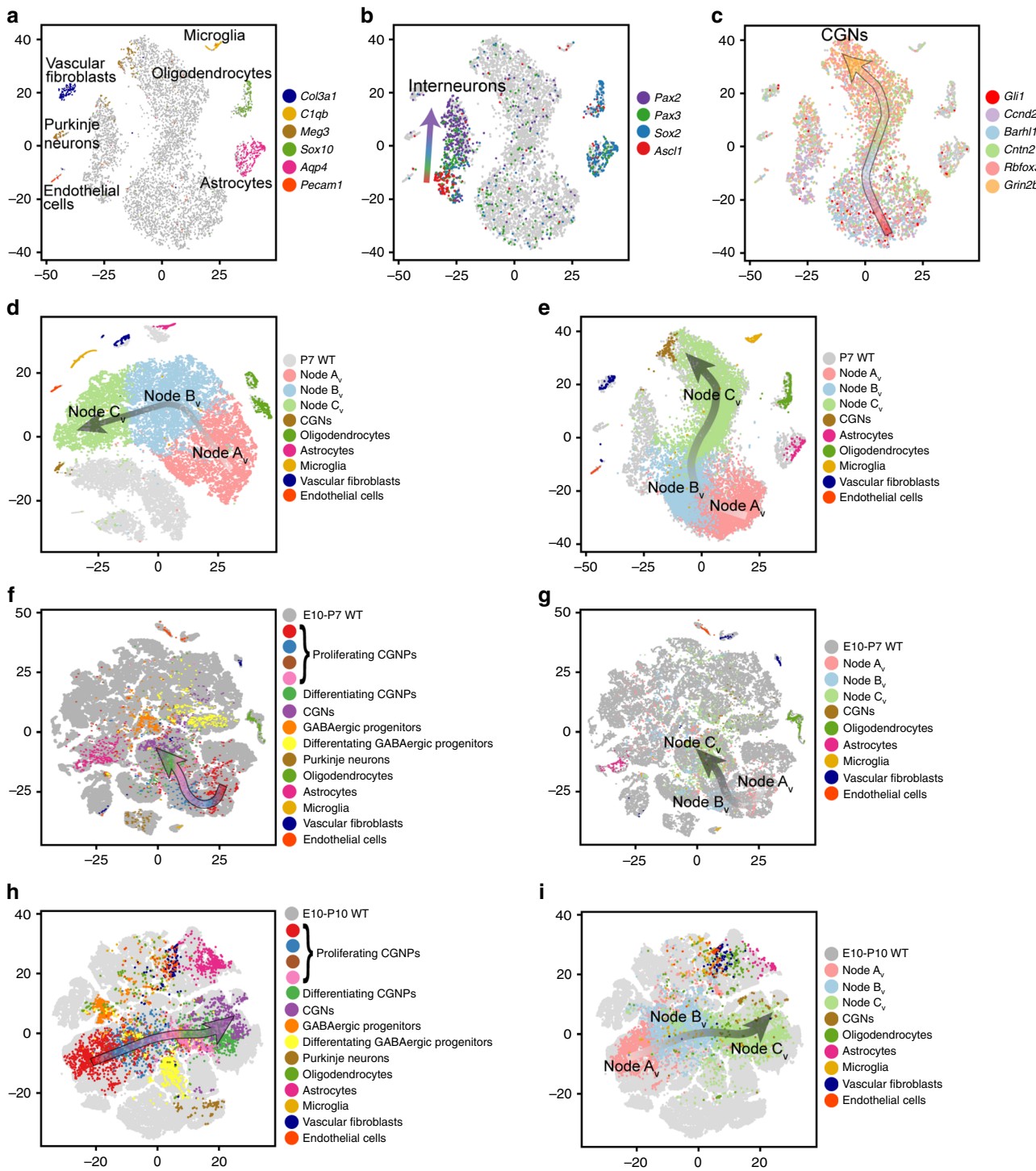

**Fig. 3 The tumor cells projected on WT P7 cerebellum predominantly map to CGNPs. a** t-SNE projection of cells from five WT P7 cerebella, based on PCA, color-coded for the expression of indicated vascular, microglial and glial markers. **b** Feature plot of *Pax2*, *Sox2*, *Pax3*, *Ascl1* on the t-SNE projection in **a**. **c** Feature plot of CGNP markers *Gli1*, *Ccnd2*, *Barhl1*, *Cntn2*, *Rbfox3*, *Grin2b* on the t-SNE projection in **a**. **d** Seurat CCA analysis of M-Smo and WT cells with co-clustering of stromal cell types and no co-clustering of Nodes $A_V$–$C_V$. **e** Cells from the five vehicle-treated M-Smo tumors, mapped using the k-NN algorithm onto the WT P7 cerebella t-SNE projection in **a**, with Nodes $A_V$–$C_V$ and identified cell types color coded. **f, h** P7 WT cells mapped by kNN onto the t-SNE of WT cells from **f** Vladoiu et al.[34] or **h** Carter et al.[34]. **g, i** Vehicle-treated M-Smo cells mapped by kNN onto the t-SNE of WT cells from **g** Vladoiu et al.[34] or **i** Carter et al.[34]. On t-SNE plots with multiple markers, individual cells may express more than one marker and markers are over-plotted in the order listed. Individual markers may be separately plotted at http://gershon-lab.med.unc.edu/single-cell/.

$A_V$–$C_V$, as defined by the PCA and HCA, again localized to discrete and adjacent regions, indicating that the types of cells identified by the PCA of all cells and the ICA of the CGNP-like tumor cells showed close correspondence (Fig. 4a).

This ICA-directed t-SNE projection suggested two discrete patterns of transition. IC $2_V$ defined a gradient of neuronal differentiation across Node $C_V$ (Fig. 4b). Cells outside this gradient were arranged in a circular pattern that corresponded

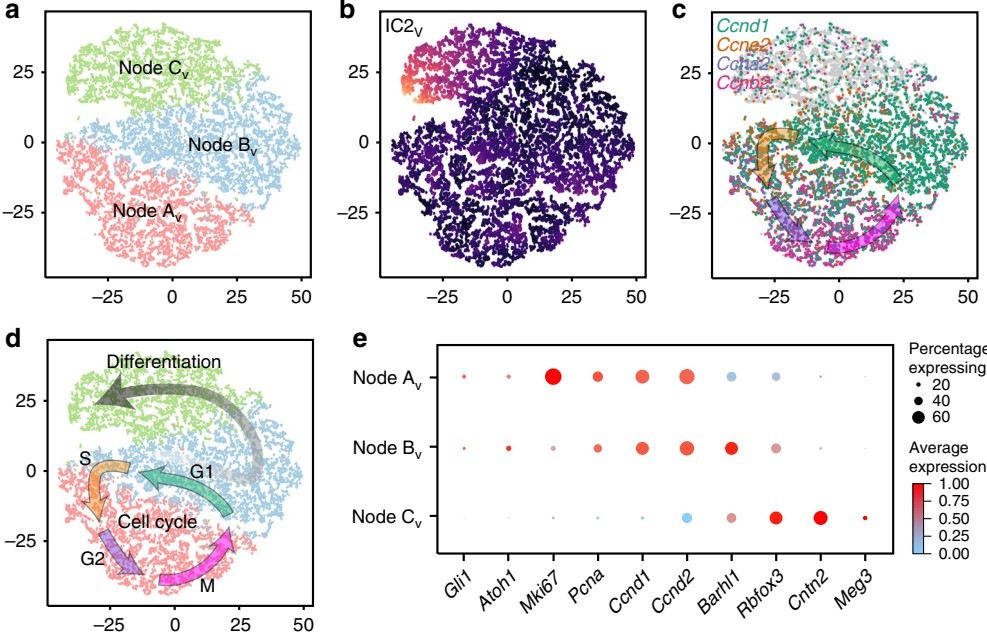

**Fig. 4 ICA on CGNP-like tumor cells demonstrates self-renewal and differentiation. a** t-SNE projection of CGNP-like cells from vehicle-treated tumors directed by ICA, with Nodes A$_V$–C$_V$ color-coded. **b** ICA t-SNE as in **a**, with IC2$_V$ color-coded from yellow (high) to blue (low). **c** Feature plot of *Ccnd1, e2, a2* and *b2* on ICA t-SNE in **a**. Arrows are placed according to indicated marker expression and show the expected sequence of Cyclins through the cell cycle. **d** Pseudotime trajectories inferred by IC2$_V$ and Cyclin expression, plotted on ICA t-SNE in **a**. **e** Dot Plot showing expression of developmental markers across Nodes A$_V$–C$_V$. Average expression values are calculated from all cells in each Node, including cells with no detectable copies of the transcript.

with the cell cycle, as demonstrated by the sequential expression of Cyclins *Ccnd1, Ccne2, Ccna2* and *Ccnb2* in adjacent regions, marking G1, S, G2 and M phases, respectively (Fig. 4c). This same progression was demonstrated by using the larger cell cycle phase-specific gene lists identified in prior single-cell transcriptomic studies[32] (Supplementary Fig. 3b–f). Node B$_V$ mapped to the point of divergence between the circular path of cycling cells and the linear path of differentiating cells. Based on these observations, we infer that ICA sorts medulloblastoma cells into various stages of two fundamental processes, the cyclic self-renewal of proliferative cells and the terminal differentiation of cells exiting the cell cycle (Fig. 4d).

The spectrum of developmental states of the medulloblastoma cells could be parsed effectively with a series of markers selected from the differential genes expressed by each node. We found sequential expression across the nodes of the proliferation markers *Mki67, Pcna, Ccnd1* and *Ccnd2*, SHH-pathway marker *Gli1*, CGNP markers *Atoh1*, and *Barhl*, late CGNP marker *Cntn2* (aka *Tag1*), and the neuronal markers *Rbfox3* (aka *NeuN*) and *Meg3* (Supplementary Fig. 4). These markers parsed the progression of cells from Nodes A$_V$–C$_V$ along a CGNP-like developmental trajectory from proliferative and undifferentiated to proliferative with early neural differentiation to non-proliferative, with advanced neural differentiation (Fig. 4e), consistent with the differentiation trajectory identified by ICA and terminating in mature neurons.

**Lineage tracing identifies tumor-derived glial cells**. The *SmoM2* transgene includes a 3′ *Yfp* sequence[27] that we used to trace tumor cell lineage. Expression of *SmoM2* requires Cre-mediated stop codon excision, which in *M-Smo* mice is limited to the descendants of cells expressing *Math1-Cre*. The expression of *Yfp* thus identified cells descended from *Math1-Cre* expressing predecessors. As expected, we found *Yfp*+ cells scattered throughout Nodes A$_V$–C$_V$ (Fig. 5a), indicating *Math1-Cre* lineage. Because high-throughput single-cell transcriptomics using Drop-Seq

captures 10–20% of mRNAs in each cell, the sensitivity of detecting individual markers may be low and varies with expression levels. The true rate of *Yfp* expression is therefore likely higher than the rate detected and it is possible that most or all of the cells in Nodes A$_V$–C$_V$ are derived from *Math1-Cre*-expressing predecessors. Inconsistent with the expected *Math1-Cre* lineage, however, we also found *Yfp*+ cells in the astrocytic and oligodendrocytic clusters. Quantification showed that 8.04 ± 2.84% (mean ± SEM) of the *Yfp*+ cells expressed the astrocytic marker *Gfap* and 7.59 ± 1.16% expressed the oligodendroglial marker *Sox10*. The co-expression of *Yfp* with *Gfap* or *Sox10* indicates that *Math1-Cre* expressing predecessors in *M-Smo* tumors give rise to progeny with glial phenotypes.

The *Yfp*+ glial cells in *M-Smo* mice suggested a lineage expansion caused by SHH hyperactivation. Extensive lineage tracing studies in normal mice show that *Math1* expression defines a set of cells with neural commitment[29,39]. However, these studies were intended to characterize the predominant fate of the *Math1*+ cells, rather than to test for the possibility of rare deviations from the typical trajectory. Moreover, *Math1-Cre* is a synthetic transgene and the domain of *Cre* expression in *Math1-Cre* mice may not exactly correspond with native *Atoh1* expression. To determine experimentally whether *Math1-Cre* predecessors in normal cerebellum give rise to cells with glial phenotype, we used flow cytometry to trace lineage in cerebella from *Math1-Cre/Pham* mice, in which *Math1-Cre* activates expression of the fluorescent reporter DENDRA2. We then quantified the co-expression of DENDRA2 and GFAP. We found that 9.53 ± 0.73% (mean ± SEM) of the DENDRA2− cells were GFAP+ compared to 0.04 ± 0.01% of the DENDRA2+ cells (Fig. 5b), indicating that GFAP+ descendants of *Math1-Cre*-expressing cells were extremely rare in cerebella without tumors.

To trace the Math1 lineage in tumors, we bred *Math1-Cre/Pham* and *SmoM2* mice. The resulting *Math1-Cre/SmoM2/Pham* (*M-Smo^Pham*) mice developed tumors with the descendants of *Math1-Cre*-expressing cells marked by DENDRA2. We

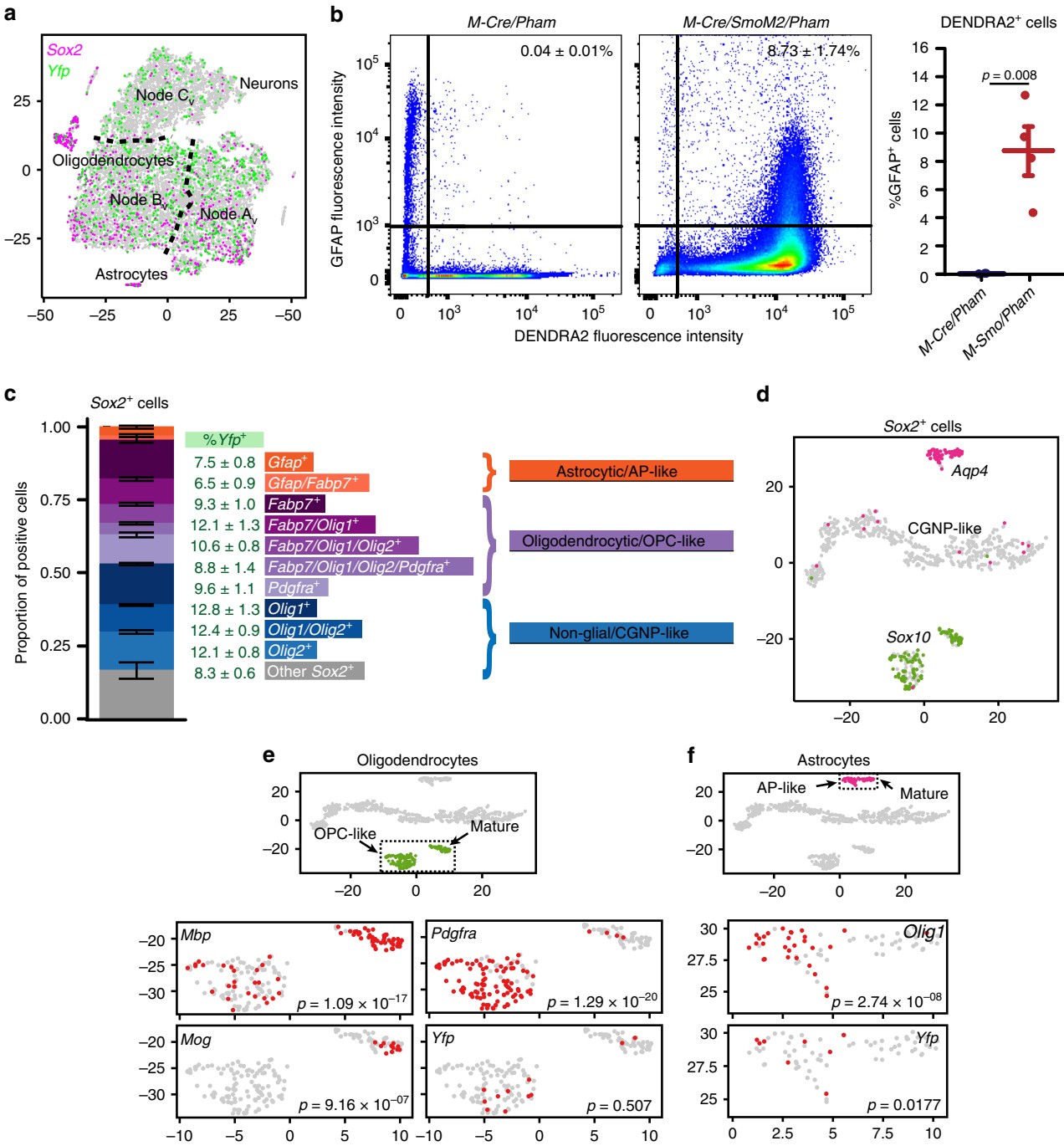

**Fig. 5 Tumors show lineage expansion and stem cell marker expression. a** Feature plot of *Sox2* and *Yfp* on the t-SNE projection of Fig. 2a. **b** Representative flow cytometry studies, plotting GFAP expression of each cell versus DENDRA2 expression, with thresholds for positivity indicated by solid lines, and with the fraction of GFAP+ cells in the DENDRA2+ cerebellar or tumor populations of replicate mice of indicated genotypes. **c** Bar plot showing the proportions of *Sox2*+ cells expressing indicated markers in each of the five vehicle-treated tumors. Error bars are SEM. **d** t-SNE projection of *Sox2*+ cells from the five vehicle-treated tumors, directed by ICA, with feature plots of *Sox10* and *Aqp4*. **e** Oligodendrocytic clusters of the t-SNE projection in **d**, with feature plots of *Pdgfra*, *Mbp1*, *Plp1*, *Mog* and *Yfp*. **f** Astrocytic clusters of the t-SNE projection in **d**, with feature plots of *Olig1* and *Yfp*. *p* Value in **b** determined by two-sided Student's *t*-test and in **e** and **f**, by Fisher's exact test.

then used flow cytometry to quantify the co-expression of DENDRA2 and GFAP. In dissociated P15 *M-Smo^{Pham}* tumors, $8.73 \pm 1.74\%$ of the DENDRA2+ cells were GFAP+ (Fig. 5b), >200-fold higher than the GFAP+ fraction of DENDRA2+ cells in P7 cerebella ($p = 0.00832$). The GFAP+ fraction in the DENDRA2+ population of *M-Smo^{Pham}* tumors closely matched the *Gfap*+ fraction of the *Yfp*+ population *M-Smo* tumors, providing an alternative confirmation of glial marker

expression by descendants of *Math1-Cre*-expressing cells within medulloblastomas.

These studies show that SmoM2-driven tumorigenesis expanded the range of fates of the *Math1-Cre* lineage. This finding is consistent with prior studies demonstrating that SHH hyperactivation in vitro can expand the *Math1* lineage, as CGNPs cultured with SHH and BMP4 give rise to astrocytes[40]. The expansion of cell fates that we observed in *M-Smo* tumors was

selective, as two-way ANOVA showed that the rate of $Yfp^+$ cells in the endothelial, vascular fibroblast and microglial clusters was significantly lower than in the astrocytic and oligodendrocytic clusters ($p < 0.001$ for each comparison). Thus, medulloblastoma cells assumed a range of fates that was broader than the expected neuronal fate of the $Atoh1$-expressing progenitors but remained within the neuroectodermal lineage.

**Tumors contain diverse cells expressing stem cell markers.** In light of the expanded fates in $Math1$-$Cre$-descended cells of $M$-$Smo$ tumors, we examined the transcriptomic data for evidence of multipotent stem-like cells. Prior studies analyzed $Sox2$-$eGFP$ expression in radiation-induced medulloblastomas that form in $Ptch^{+/-}/Sox2$-$eGFP$ mice to identify stem-like cells, whose properties included self-renewal, tumor recapitulation on transplantation, and the ability to give rise to progeny with diverse differentiation[31]. Bulk transcriptomic analysis of eGFP+ tumor cells from these mice showed upregulation of specific stem cell markers, including $Sox2$, $Gfap$, $Olig1$, $Olig2$, $Blbp$ ($Fabp7$) and $Pdgfra$[31].

We detected $Sox2^+$ cells in our single-cell transcriptomic data from $M$-$Smo$ tumors. The set of genes upregulated in these cells compared to all other cells (Supplementary Dataset 4) resembled the transcriptomic profile of Sox2-eGFP+ cells identified by Vanner et al.[31] (hypergeometric test, $p = 3.2 \times 10^{-124}$). However, the $Sox2^+$ population was diverse, with individual $Sox2^+$ cells mapping to different regions of the t-SNE, including the astrocytic and oligodendrocytic clusters and the CGNP-like cells of Nodes $A_V$–$C_V$ (Fig. 5a). Stem cell markers identified in Vanner et al.[31] were not evenly expressed across the $Sox2^+$ population; rather, different subsets of $Sox2^+$ cells expressed different combinations of the markers (Fig. 5c). Not all $Sox2^+$ cells were tumor-derived, as glial cells in normal P7 cerebellum also expressed $Sox2$ (Fig. 3b). Altogether, the $Sox2^+$ population comprised diverse cell types including both normal glia and tumor-derived cells with either glial and neural progenitor-like characteristics.

To resolve the different types of $Sox2$-expressing cells, we subjected the $Sox2^+$ cells from the vehicle-treated tumors to ICA and Louvain clustering. The resulting t-SNE projection defined three discrete groups, a $Sox10^+$ oligodendrocytic group, an $Aqp4^+$ astrocytic group and a third group composed of CGNP-like cells from Nodes $A_V$–$C_V$ (Fig. 5d). The oligodendrocytic and astrocytic groups both included two discrete subgroups with differential gene expression (Supplementary Dataset 5). One oligodendrocytic subgroup was $Pdgfra^+$ and proliferative, as demonstrated by $Ccnd1$ expression, while the other showed markers of differentiation, including myelin genes $Mbp1$ and $Mog$ (Fig. 5e). We considered the $Pdgfra^+$ group to be the oligodendrocyte precursors (OPCs). The two astrocytic subgroups were differentiated by the absence or presence of $Olig1$ (Fig. 5e). We considered the $Olig1^+$ subset to be the astrocyte precursors (APs). Within the astrocytic group, $Yfp^+$ cells distributed entirely within the AP subset ($p = 0.0177$, Fisher's exact test), while within the oligodendrocytic group, $Yfp^+$ cells distributed across both groups (Fig. 5e). Sox2+ cells thus included glial and neural-progenitor-like subsets that varied in differentiation, and tumor-derived $Sox2^+$ astrocytic cells tended to be undifferentiated.

**Inhibiting SHH signaling at SMO promotes differentiation.** To determine the effect of SHH inhibition on cellular heterogeneity and developmental trajectory, we compared single-cell gene expression data from vismodegib-treated and vehicle-treated tumors, including >30,000 cells harvested from a total of ten

tumors. We identified PCs using the same workflow as in the vehicle-only and WT analyses and used these PCs for t-SNE visualization and Louvain clustering.

We again found that cells localized in the t-SNE in several single-cluster groups and in one large multi-cluster group. Specific markers identified the single-cluster groups as endothelial cells, vascular fibroblasts, microglia, neurons, oligodendrocytes and astrocytes (Fig. 6a). The multi-cluster group comprised 11 clusters which hierarchical cluster analysis divided into four nodes, designated $A_{Together}$–$D_{Together}$ (Nodes $A_T$–$D_T$; Fig. 6b). Nodes $A_T$–$C_T$ showed phenotypes that matched Nodes $A_V$–$C_V$, while Node $D_T$ comprised cells with increased expression of late differentiation markers (Fig. 6c). A small portion of the vehicle-treated cells that populated Node $C_V$ were placed in Node $D_T$, however >85% of Node $D_T$ derived from vismodegib-treated tumors (Fig. 6d; $p = 0.007$ by two-sample $T$-test), indicating that vismodegib induced a more differentiated state. Cells throughout Node $D_T$ expressed $Yfp$, confirming the tumor lineage of these differentiated cells (Fig. 6e).

Increased differentiation was further demonstrated by mapping the cells of vehicle- and vismodegib-treated tumors according to their best fit among the cells of the WT cerebellum using kNN. Compared to vehicle-treated tumors, the CGNP-like cells from vismodegib-treated tumors mapped to more differentiated regions of the WT CGNPs in both our P7 dataset and in the datasets of Carter et al. and Vladoiu et al.[38,39], (Fig. 6f, Supplementary Fig. 5). ICA on Nodes $A_T$–$D_T$ demonstrated that vismodegib induced progression along a differentiation trajectory, with more vismodegib-treated cells at the extreme of the IC1 (Fig. 6g; Supplementary Dataset 6). Developmental mapping and ICA both confirm a shift in the CGNP-like tumor cells toward more differentiated states.

**Variation in vismodegib sensitivity.** To determine the responses of each cell type to vismodegib, we compared the populations of each node and cluster in vehicle-treated and vismodegib-treated tumors. We determined for each individual animal the number of cells in each node and cluster, normalized to the total number of cells from that animal. We then compared the distribution of cells from vismodegib-treated and vehicle-treated mice across nodes and clusters (Fig. 6h, Supplementary Fig. 6a). These comparisons showed a shift in which the proliferative populations (Nodes $A_T$ + $B_T$) were depleted by vismodegib and the non-proliferative populations (Nodes $C_T$ + $D_T$) were enriched ($p < 0.001$ by ANOVA).

The enrichment of differentiated cell types was relative, rather than absolute. Vanner et al.[31] previously showed that in medulloblastomas, differentiated progeny of stem cells tend to undergo apoptosis after a period of days[31]. Consistent with this finding, we previously found large accumulations of neurons in apoptosis-deficient medulloblastomas in $Bax$-deleted mice, suggesting that differentiating tumor cells are typically removed by BAX-regulated cell death[41]. We did not observe accumulation of neurons in medulloblastomas of $M$-$Smo$ mice after 3 weeks of vismodegib treatment (Supplementary Fig. 6b), suggesting that cell death may follow differentiation. However, we also were unable to detect a significant increase in cells showing the apoptotic marker cleaved caspase-3 (Supplementary Fig. 6c), consistent with either no increase in cell death, or a small, asynchronous increase that is hard to detect. Further studies with $Bax$-mutant $M$-$Smo$ mice may be needed to determine if vismodegib-induced differentiation is followed by latent cell death.

Depletion of proliferating cell types by vismodegib was not uniform. The population of Node $B_T$ showed a statistically significant ($p < 0.001$) 2-fold decrease. In contrast, the population

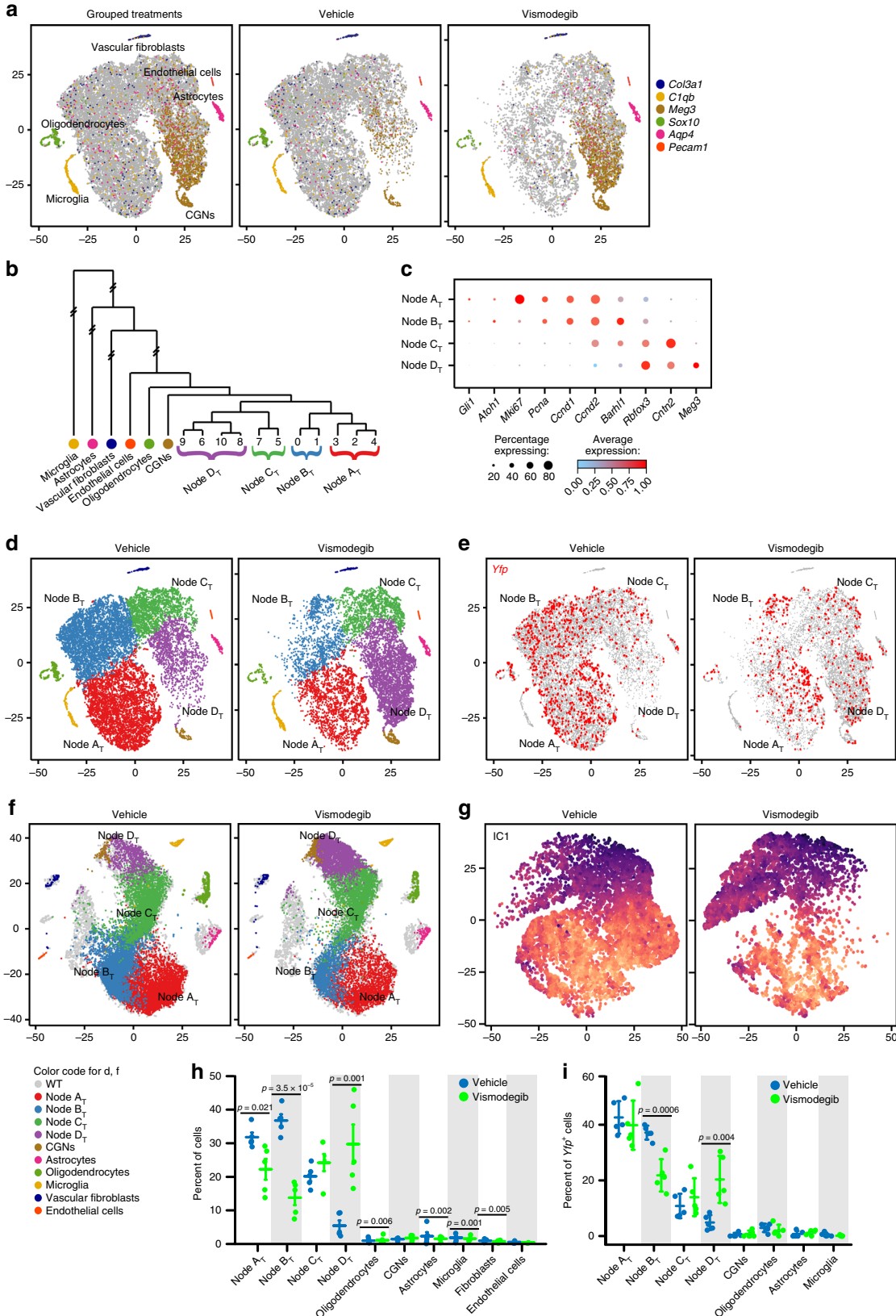

of Node $A_T$ showed a 30% decrease that was not statistically significant. Similar patterns were also seen in the $Yfp^+$ subset of cells, with statistically significant depletion of $Yfp^+$ cells in Node $B_T$, statistically significant enrichment of $Yfp^+$ cells in Node $D_T$ and no statistically significant changes in the $Yfp^+$ fractions of

Node $A_T$ or the astrocytic or oligodendroglial clusters (Fig. 6i). The different effects of vismodegib on the node populations identify the proliferating cells of Node $B_T$ as vismodegib-sensitive and the proliferating cells of Node $A_T$, along with the $Yfp^+$ glial cells, as relatively vismodegib-resistant.

**Fig. 6 Vismodegib advances differentiation and depletes subsets of tumor cells. a** t-SNE projection of cells from five vehicle-treated and five vismodegib-treated tumors, directed by PCA, plotted either with cells from both conditions shown together, or in separate images, and color-coded for the expression of indicated markers. **b** Dendrogram depicts the HCA that identified Nodes $A_V$–$C_V$. **c** Dot Plot showing expression of developmental markers across Nodes $A_V$–$C_V$. **d** Nodes and identified cell types color-coded on t-SNE from **a**. **e** Feature plot of *YFP* on the t-SNE from **a**. **f** Cells from the five vehicle-treated or five vismodegib-treated tumors, mapped using the k-NN algorithm onto the t-SNE projection of WT P7 cerebella (Fig. 3a), with Nodes $A_T$–$D_T$ and identified cell types color-coded. **g** ICA-directed t-SNE projection of CGNP-like cells from vehicle-treated and or vismodegib-treated tumors, with the differentiation IC color-coded. **h** Fractional population of indicated groups from vehicle-treated and vismodegib-treated mice, normalized to the total number of cells per mouse. Each dot represents an individual replicate animal. Horizontal lines indicate the means, and error bars indicate SEM. **i** Fractional population changes in *Yfp*+ cells induced by vismodegib, formatted as in **g**. *p* Values in **h** and **i** determined by one-way ANOVA.

**Persistent SHH activation in vismodegib-resistant cells**. We found that SHH activation persisted in the cells that remained in proliferative Node $A_T$ in vismodegib-treated tumors, demonstrated by continued expression of the SHH pathway markers *Gli1* (Fig. 7a), *Ptch1*, *Hhip* and *Sfrp1* (Supplementary Fig. 7). These results were consistent with the resumption of Gli-luc signal after 3 days of vismodegib treatment (Fig. 1e) and indicate that in a subset of cells, vismodegib failed to suppress SHH-driven transcription. We hypothesized that this failure could potentially arise from a pharmacodynamic mechanism in which the drug is present but unable to block the pathway, or from a pharmacokinetic mechanism, in which the unaffected cells are not exposed to the drug, due to local variation in drug penetration.

To test for uneven distribution of vismodegib, we visualized drug distribution using mass spectrometry imaging (MSI) accomplished by infrared matrix-assisted laser desorption electrospray ionization (IR-MALDESI). For this purpose, brains were harvested from vismodegib-injected and control mice, rapidly frozen, and sectioned in the sagittal plane. The frozen sections were then scanned progressively across the section by IR-infrared laser and the resulting ionized species arising from each successively scanned region were detected by mass spectrometry, generating concentration maps for each ion across the entire section. We were able to map vismodegib, and to compare vismodegib concentration to endogenous metabolites. We found cholesterol in relatively high concentrations throughout the brain but low concentrations in all regions of the tumor (Fig. 7b). In contrast to cholesterol, vismodegib was evenly distributed across the brain and tumor, without local variation (Fig. 7b). This even distribution argues against a local pharmacokinetic cause for differential effects of vismodegib.

**HES1 marks vismodegib-sensitive tumor cells**. To gain insight in the mechanisms of differential pharmacodynamic response to vismodegib in different cell types, we compared gene expression in untreated cells of the vismodegib-sensitive and vismodegib-resistant clusters. Noting that cluster 0 of Node $B_T$ was most strongly depleted by vismodegib (Supplementary Fig. 6a), we generated a list of genes differentially upregulated in Cluster 0 in vehicle-treated tumors compared to all other vehicle-treated cells (Supplementary Dataset 7). Focusing on the transcription factors on this list, we noted specific expression of the Notch pathway transcription factor *Hes1*. Analysis of *Hes1* expression in the vehicle-treated subset of the t-SNE showed that in control tumors *Hes1*+ cells predominantly localized to cluster 0 in Node $B_T$, but were also found in clusters within Node $A_T$, and in the astrocytic and oligodendrocytic clusters (Fig. 7c). The *Hes1*+ cells in vehicle-treated tumors were predominantly *Gli1*+, indicating SHH-pathway activation. Vismodegib decreased *Hes1* expression in Nodes $A_T$ and $B_T$, while glial clusters remained *Hes1*+ (Fig. 7c, g top panel).

We analyzed HES1 protein expression in tumor sections by immunohistochemistry to confirm these observations and to determine the spatial distribution and proliferative state of HES1+ cells. HES1+ cells were distributed throughout the vehicle-treated tumors (Fig. 7d) and co-localized with pRB, consistent with the proliferative phenotype predicted by the transcriptomic data. Vismodegib reduced pRB in tumors, reduced HES1 expression and markedly reduced pRB in the HES1+ population (Fig. 7g middle panel). To determine if the reduction in HES1+ cells persisted over time, we examined HES1 expression in tumors in M-Smo mice with vismodegib for 14 days. In these tumors, HES1+ cells were as rare as in tumors treated for 3 days, and pRB was significantly reduced in the HES1+ population (Supplementary Fig. 8a). These data show that proliferative HES1+ tumor cells were widely distributed and disproportionately inhibited by vismodegib.

**MYOD1 marks vismodegib-resistant tumor cells**. In contrast to *Hes1*, the transcription factor *Myod1* identified a subset of tumor cells that remained proliferative after vismodegib treatment. We identified *Myod1* by searching for transcription factors in the set of genes upregulated in vismodegib-treated cells of Node $A_T$ compared to vehicle-treated cells of Node $B_T$ (Supplementary Dataset 8). Analysis of *Myod1* expression in the vehicle subset of the t-SNE showed that *Myod1*+ cells localized to the *Gli1*+ Nodes $A_T$ and $B_T$ (Fig. 7a, e). After vismodegib treatment, cells in Nodes $A_T$ and $B_T$ continued to express *Myod1*, and these *Myod1*+ cells continued to express *Gli1* (Fig. 7a, e). Analysis of vehicle-treated tumor sections demonstrated MYOD1-expressing cells throughout the tumors. (Fig. 7f). Dual labeling in sections (Fig. 7f, g) and flow cytometry showed that these cells were pRB+ and distributed across the cell cycle (Fig. 7h), confirming the proliferative state predicted by the transcriptomic data. Vismodegib produced a trend toward reduced MYOD1+ cells that was not statistically significant (Fig. 7g, h), did not significantly reduce pRB expression in the MYOD1+ population, and only modestly altered cell cycle phase distribution (Fig. 7f–h). In contrast to HES1+ cells, MYOD1+ cells continued to proliferate in tumors treated for 14 days, showing a statistically significant decrease in relative fraction but no decrease in pRB+ fraction (Supplementary Fig. 8b). These data show that HES1 and MYOD1 mark subsets of proliferating tumor cells with markedly different sensitivities to vismodegib.

**HES1 and MYOD1 in clinical medulloblastoma samples**. We analyzed transcriptomic data from human medulloblastomas to determine whether *HES1* and *MYOD1* are frequently expressed. We found heterogeneous expression of both genes in published data[42] from patient-derived tumor samples (Fig. 7i, j; Supplementary Fig. 9). We noted variation in mean expression across medulloblastoma subtypes, and wide variability within each subtype. These data show that human tumors contain *HES1*+ and *MYOD1*+ cells, and that in human tumors, as in mouse tumors, expression of these genes is heterogeneous.

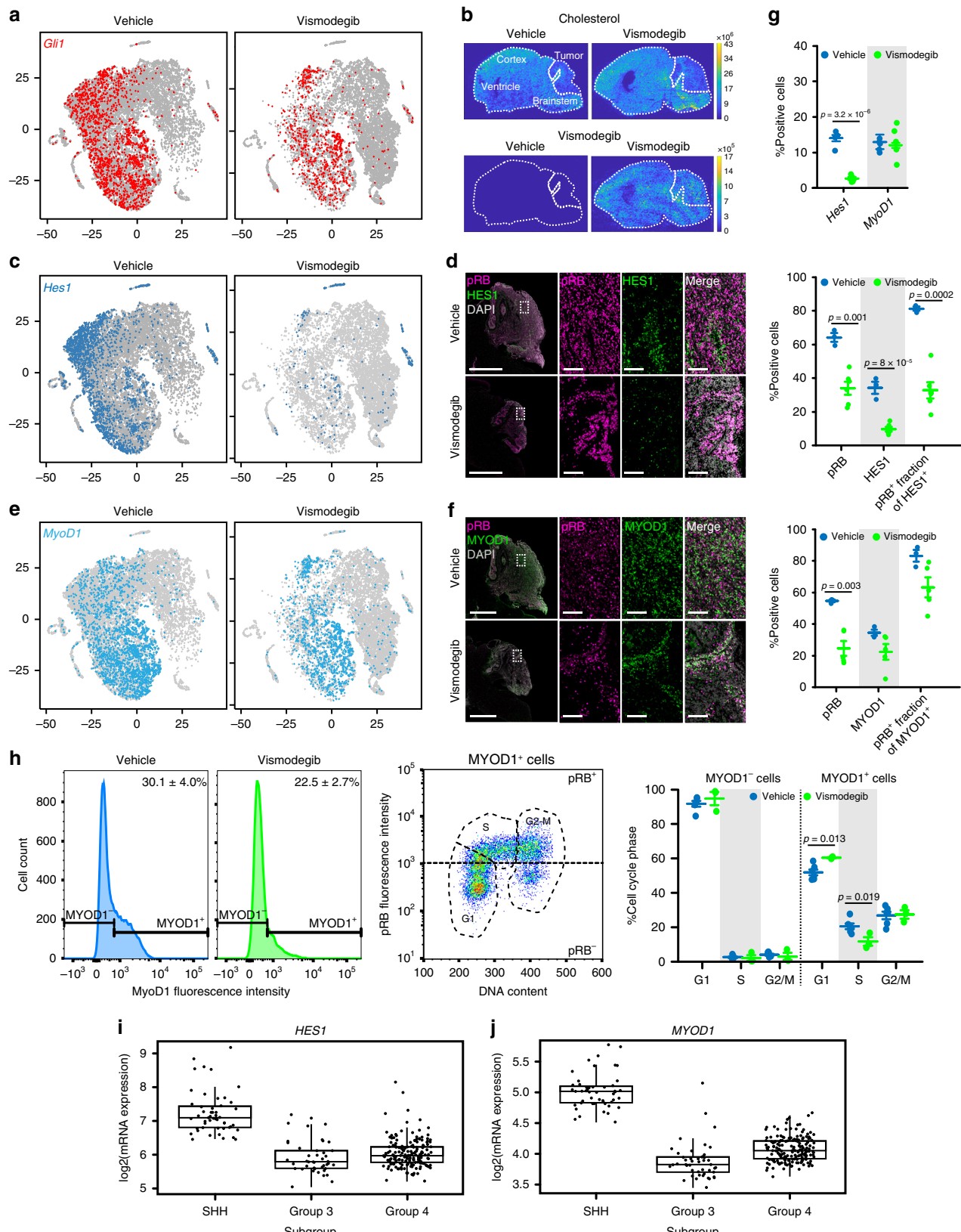

**Reduced Sufu in vismodegib-resistant tumor cells**. Mutations of intracellular SHH-pathway regulator *Sufu* can confer vismodegib resistance[25] and we therefore analyzed the effect of Sufu expression on cellular response to treatment. We plotted *Gli1* expression as a function of *Sufu* expression, and for comparison plotted the SHH target *Ptch1* as a function of *Gli1*. To prevent cells with

stochastic drop out of *Gli1*, *Ptch1* or *Sufu* from biasing the data, we used the Markov affinity-based graph imputation of cells (MAGIC) algorithm[43] to impute expression values in cells where 0 transcripts of each gene were detected. For *Ptch1*, this method showed a statistically significant positive correlation with *Gli1* in both vehicle-treated and vismodegib-treated tumors ($r = 0.971$,

**Fig. 7 HES1 and MYOD1 mark vismodegib-responsive and resistant subsets of cells. a** t-SNE projections as in Fig. 6a, with *Gli1*+ cells indicated.
**b** MALDESI images of cholesterol and vismodegib distribution in sagittal brain sections including both tumor and normal brain. **c** t-SNE projections as in
**a** with *Hes1*+ cells indicated. **d** IHC for HES1 and pRB in sagittal hindbrain sections from representative *M-Smo* mice treated as indicated. **e** t-SNE projections
as in **a**, with *Myod1*+ cells indicated. **f** IHC for MYOD1 and pRB in sagittal hindbrain sections from representative *M-Smo* mice treated as indicated.
**g** Quantification of data from **c–f**. Dots represent individual replicate mice. Horizontal lines represent means and error bars indicate SEM. **h** Analysis of
MYOD1, pRB and cell cycle phase by flow cytometry. Graphs formatted as in **g**. **i**, **j** Expression microarray data on *HES1* and *MYOD1* expression in human
medulloblastoma samples, as presented in ref. [44], with boxes indicating the mean, 25th and 75th percentile and whiskers indicating the 10th and 90th
percentiles. Scale bars = 2 mm, except in insets where scale bars = 100 μm. *p* Values determined by two-sided Student's *t*-test.

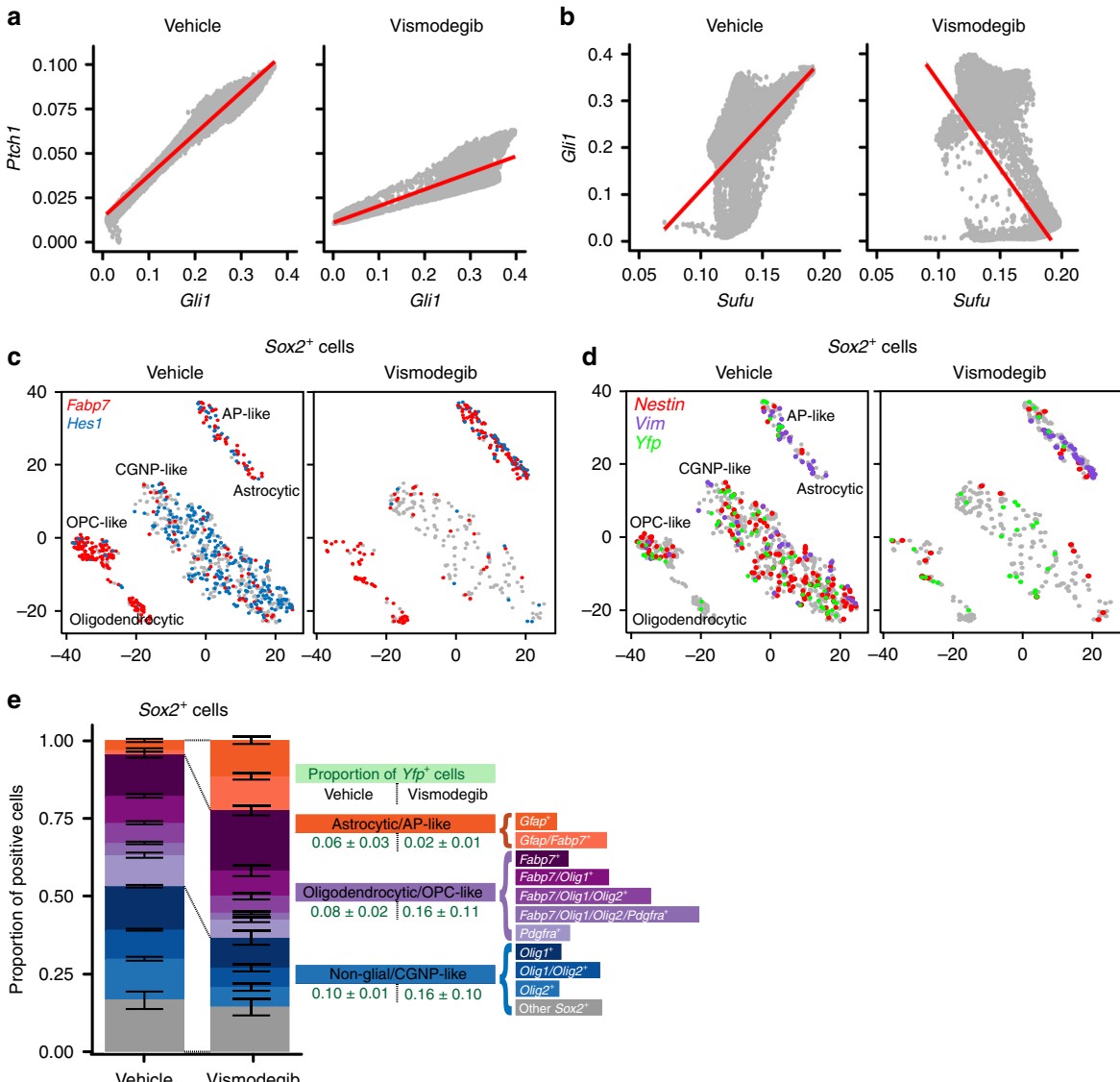

**Fig. 8 Factors that maintain proliferative and stem-like phenotypes after vismodegib. a** Correlation plot after imputation of data, showing expression of
*Gli1* on the X-axis and *Ptch1* on the Y-axis, in vehicle- and vismodegib-treated tumors graphed separately. Each dot represents an individual cell.
**b** Correlation plot after imputation of data, showing expression of *Sufu* on the X-axis and *Gli1* on the Y-axis, in vehicle and vismodegib-treated tumors
graphed separately as in **a**. **c** t-SNE projection of *Sox2*+ cells from vehicle-treated and vismodegib-treated tumors, directed by ICA with expression of *Fabp7*
and *Hes1* color coded. **d** t-SNE projection of *Sox2*+ cells from the five vehicle-treated and five vismodegib-treated tumors, directed by ICA, with expression
of *Nes*, *Vim* and *Yfp* indicated. **e** Bar plots comparing the proportions of *Sox2*+ cells expressing indicated markers in each of the five vehicle-treated or
vismodegib-treated tumors. Error bars are SEM. The *Yfp*+ fractions are indicated.

$p < 1 \times 10^{-15}$ and $r = 0.925$, $p < 1 \times 10^{-15}$, respectively; hypothesis testing of correlation by t-distribution; Fig. 8b). In contrast, *Sufu* showed a statistically significant positive correlation with *Gli1* in vehicle-treated tumors ($r = 0.406$, $p < 1 \times 10^{-15}$; hypothesis testing of correlation by t-distribution), but a statistically

significant negative correlation in vismodegib-treated tumors ($r = -0.566$, $p < 1 \times 10^{-15}$; hypothesis testing of correlation by t-distribution; Fig. 8c). This negative correlation persisted when Node $D_T$ cells, which were rare in the vehicle-treated tumors and numerous in the vismodegib-treated tumors, were removed from

the analysis (Supplementary Fig. 10). Further, we used ANCOVA to test whether the relationship between *Sufu* and *Gli1* was dependent on treatment ($p < 1 \times 10^{-16}$; ANCOVA Gli1 ~ Sufu*Treatment). Based on this statistically significant effect of treatment, which induced a negative correlation between *Sufu* and *Gli*, we propose that that vismodegib may sensitize medulloblastoma cells to the inhibitory effect of SUFU, such that cells with higher SUFU differentiate while cells with lower SUFU remain proliferative.

**Sox2$^+$ cells respond heterogeneously to vismodegib.** Single-cell analysis identified subsets of cells with variable sensitivity to vismodegib within the *Sox2$^+$* population. To analyze the effect of vismodegib on the heterogeneity within this population, we divided the *Sox2$^+$* population into a glial population that was *Fabp7$^+$*, and a CGNP-like population that was *Fabp7$^-$*. ICA-directed analysis of the *Sox2$^+$* populations of vehicle-treated and vismodegib-treated tumors showed that the vehicle-treated tumors contained a significant population of CGNP-like *Sox2$^+$* cells that were *Hes1$^+$* AND Fabp7$^-$ (41.1 ± 2.5%; five replicates) and that this population was specifically depleted by vismodegib (5.1 ± 2.9%; five replicates; FC = 0.1245, $p = 1.5 \times 10^{-5}$; Student's *t*-test; Fig. 8d, e). In contrast, the oligodendrocytic, OPC-like, astrocytic and AP-like subsets of *Sox2$^+$* cells were relatively preserved in vismodegib-treated tumors (Fig. 8d, e). Comparing *Sox2$^+$* cells in vehicle-treated and vismodegib-treated tumors and stratifying using *Yfp* expression and the markers identified by Vanner et al.[31], we identified subsets that were relatively depleted or enriched by vismodegib treatment, including *Sox2$^+$* cells of tumor lineage (Fig. 8f). Thus, within the *Sox2$^+$* population, as with the broader tumor population, specific subsets of cells were relatively sensitive or resistant to vismodegib, and tumor-derived cells with stem cell-like transcriptomes persisted after vismodegib treatment.

## Discussion

We found that medulloblastomas contain diverse types of tumor cells, including CGNP-like cells in a spectrum of differentiation states, and tumor-derived cells with patterns of gene expression typical of astrocytic precursors and oligodendrocytic precursors, fates outside the expected *Atoh1* lineage of the CGNPs. The expression patterns of all genes detected in our studies in the P7 WT cerebella and in treated and control tumors can be plotted through our web-based application, available at gershon-lab.med.unc.edu/single-cell/. In both the recapitulation of expected developmental trajectories and the presence of neoplastic OPC-like and AP-like cells, the *M-Smo* medulloblastomas resembled other malignant brain tumors that have been similarly subjected to single-cell transcriptomic analysis[6,9]. While vismodegib produced an overall increase in differentiation, different populations of tumor cells demonstrated different responses to disruption of the SHH pathway. Vismodegib treatment eliminated a *Hes1$^+$* subset of proliferating, CGNP-like cells, while a *Myod1$^+$* subset remained proliferative, and continued to show SHH-pathway activation. Within the *Sox2$^+$* population, we noted a similarly heterogeneous response to SHH inhibition, with *Fabp7$^-$/Hes1$^+$/Sox2$^+$* cells depleted by vismodegib and OPC-like, AP-like and *FABP7$^-$/Myod1$^+$/Sox2$^+$* populations persisting. The diversity of vismodegib-resistant cell types identifies several populations of cells that may drive recurrence and treatment failure.

The identification of subgroups within the *Sox2$^+$* population, including vismodegib-resistant *Sox2$^+$* stem-like cell types, extends the prior finding that *Sox2$^+$* cells are tumor propagating cells that can repopulate tumors after vismodegib therapy[31]. Additionally, we found populations of vismodegib-resistant cells

that express *Myod1*, of which only a fraction express stem cell markers. MYOD1$^+$ cells continued to express pRB after continued administration of vismodegib, demonstrating continued propagation during therapy. Thus at the outset of treatment, before a long period of selective pressure, *M-Smo* tumors already contain several populations of cells that no longer depend on the initiating oncogenic mutation.

The expression of *Hes1* in CGNP-like cells of Nodes A$_T$ and B$_T$ may reflect active NOTCH pathway signaling, or alternatively direct induction of *Hes1* by SHH pathway hyperactivation. *Hes1* is known to be a target of NOTCH, and signaling specifically through NOTCH2 has been shown in both CGNP development[44] and in medulloblastoma[45]. However, prior reports also describe direct activation of *Hes1* by SHH signaling through SMO and GLI2, which binds to the *Hes1* promoter[46,47]. Consistent with a direct mechanism, SMOM2-mediated SHH hyperactivation clearly correlated with increased *Hes1* expression, as *Hes1$^+$* cells were markedly enriched in *M-Smo* medulloblastomas compared to WT P7 cerebella. The rapid depletion of the non-glial *Hes1$^+$* populations after vismodegib treatment would also support a direct relationship between SMOM2-mediated SHH hyperactivation and *Hes1* expression in tumor cells. Direct stimulation of *Hes1* by SHH would be clinically relevant as it would limit the potential for NOTCH-pathway inhibitors to alter medulloblastoma growth.

We found MYOD1 mRNA in all four medulloblastoma subtypes, and *Myod1* expression has previously been reported in both rare cells within the CGNP population, and in proliferative tumor cells in SHH-driven medulloblastoma in mice[48]. Interestingly, *Myod1* haploinsufficiency increased SHH-driven tumorigenesis in the *SmoA1* and *SmoA2* mouse models, suggesting a tumor suppressive role[48]. Such a role would not preclude the possibility that MYOD1$^+$ tumor cells drive recurrence, as MYOD1-expressing cells may have mechanisms that overcome a putative tumor-suppressive effect. Clearly, MYOD1$^+$ cells in *M-Smo* tumors are able to proliferate.

The potential lineage relationship between the *Myod1*-expressing CGNP-like cells and the *Sox2*-expressing stem-like cells requires further study. A finding that *Sox2$^+$* cells give rise to *Myod1$^+$* cells would support a hierarchical relationship. In contrast, if *Myod1$^+$* cells give rise to *Sox2$^+$* cells, that result would show that movement through the hierarchy can be bi-directional, and that pluripotency is not hierarchically limited. The observance of *Yfp$^+$* glial cells provides evidence that committed *Atoh1*-expressing progenitors re-acquired neural stem cell-like pluripotency as a consequence of SMOM2-mediated oncogenic transformation. Whether pluripotency can be increased at any time to regenerate tumor stem cells, however, remains to be answered. These lineage studies would inform the question of which subsets of cells must be targeted to block recurrence.

Our finding of persistent SHH pathway activation in the *Myod1$^+$* cells in vismodegib-treated tumors demonstrates is SMO-independent SHH activation. The inhibition of *Gli1* by vismodegib in the *Hes1$^+$* cells indicates that the SMOM2 protein is vismodegib-sensitive. The even distributions of both *Myod1$^+$* cells and vismodegib throughout the tumors argue against a role of local variation in drug levels as the cause of vismodegib failure in these cells. Rather, our data support a variation in the pharmacodynamic effect of vismodegib in different types of cells. We propose that this variation may be mediated by native SUFU, which has been related to vismodegib resistance in patients with medulloblastoma[25] and basal cell carcinoma[49]. Based on the inverse correlation between the expression of *Sufu* and *Gli1*, specifically after vismodegib treatment, we propose that vismodegib sensitizes tumor cells to SUFU-mediated inhibition by

reducing stimulation through SMO and SMOM2. Under conditions of vismodegib-mediated SMO and SMOM2 inhibition, cells with more SUFU reduce Gli1 expression and differentiate, while cells with less SUFU continue SHH-driven proliferation. We speculate that epigenetic modulation may reduce *Sufu* expression and thus prevent differentiation in the *Myod1*+ population. Targeted disruption of epigenetic modulators may clarify this mechanism, and if confirmatory, may provide a mechanism for blocking vismodegib resistance.

## Methods

Key materials including primers and antibodies, with dilutions used:

| Reagent | Concentration | Source | Identifier |
|---|---|---|---|
| **Animal studies** | | | |
| *C57BL/6* mice | N/A | The Jackson Laboratory | Stock #000664 |
| *Math1-Cre* mice | N/A | The Jackson Laboratory | Stock #011104 |
| *SmoM2-eYFP*<sup>loxP/loxP</sup> mice | N/A | The Jackson Laboratory | Stock #005130 |
| *Pham*<sup>loxP/loxP</sup> mice | N/A | The Jackson Laboratory | Stock #018385 |
| *Gli-luc* mice | N/A | Generously shared by Dr. Oren Becher, Northwestern University and Dr. Eric Holland, Fred Hutchinson Cancer Research Center | MGI #4820828 |
| *N*-Methyl-2-pyrrolidone (NMP) | 1:10 | Millipore Sigma | Catalog #328634 |
| Polyethylene glycol (PEG) | 9:10 | Millipore Sigma | Catalog #P3015-250G |
| Vismodegib | 75 mg/kg or 100 mg/kg in NMP:PEG | LC Laboratories | Catalog #V-4050 |
| Isoflurane | Vapor | Piramal Critical Care, Inc. | NCD Code #66794-017-25 |
| **PCR** | | | |
| Tail lysis buffer | 1× | Allele Biotechnology | Catalog #ABP-PP-MT01 |
| *Cre* forward primer: GCG GTC TGG CAG TAA AAA CTA TC | 200 μM | Invitrogen | JAX #oIMR1084 |
| *Cre* reverse primer: GTG AAA CAG CAT TGC TGT CAC TT | 200 μM | Invitrogen | JAX #oIMR1085 |
| *SmoM2 (YFP)* forward primer: AAG TTC ATC TGC ACC ACC G | 400 μM | Invitrogen | JAX #oIMR0872 |
| *SmoM2 (YFP)* reverse primer: TCC TTG AAG AAG ATG GTG CG | 400 μM | Invitrogen | JAX #oIMR1416 |
| *Pham* Common primer: CCA AAG TCG CTC TGA GTT GTT ATC | 200 μM | Invitrogen | JAX #13840 |
| *Pham* WT reverse primer: GAG CGG GAG AAA TGG ATA TG | 200 μM | Invitrogen | JAX #13841 |
| *Pham* Mutant reverse primer: TCA ATG GGC GGG GGT CGT T | 200 μM | Invitrogen | JAX #oIMR7320 |
| *Gli-luc* forward primer: TATCATGGATTCTAAAACGG | 200 μM | Invitrogen | N/A |
| *Gli-luc* reverse primer: CAGCTCTTT CTTCAAATCTATAC | 200 μM | Invitrogen | N/A |
| Apex Taq DNA Polymerase Master Mix | 1× | Genessee Scientific | Catalog #42-138 |
| Platinum Blue PCR SuperMix | 1× | Invitrogen | Catalog # 12580015 |
| Molecular biology grade water | – | Corning Inc. | 46-000-CM |
| **Immunofluorescence and immunohistochemistry** | | | |
| Paraformaldehyde (PFA) | 4% in PBS | | |
| Phosphate-buffered saline (PBS) | 1× | | |
| Antigen retrieval | 1:100 | Vector Laboratories | Catalog #H-3300 |
| Donkey serum | 2% in 0.3% PBST | Millipore Sigma | Catalog #D9663 |
| DAPI | 1:2500 | Invitrogen | Catalog #D1306 |
| SOX2 | 1:200 | Cell Signaling Technology | Catalog #4900 |
| HES1 | 1:200 | Cell Signaling Technology | Catalog #11988 |
| MyoD1 | 1:100 (Mouse) | Novus | Catalog #NBP2-32882-0.1 mg |
| phospho-RB (Ser807/811) | 1:3000 | Cell Signaling Technology | Catalog #8516 |
| GFAP | 1:2000 | Dako | Catalog #Z0334 |
| NeuN | 1:10,000 | Millipore | Catalog #MAB377 |
| Calbindin | 1:400 | Cell Signaling Technology | Catalog #2173 |
| Goat anti-rabbit Alexa Fluor 488 | 1:400 | Thermo Fisher Scientific | Catalog #A-11034 |
| Goat anti-mouse Alexa Fluor 555 | 1:400 | Thermo Fisher Scientific | Catalog #A-21424 |
| Novolink Polymer | Per manufacturer's instructions | Leica Biosystems | Catalog #RE7200-CE |
| ImmPRESS™ HRP Anti-Rabbit IgG | Per manufacturer's instructions | Vector Laboratories | Catalog #MP-7401 |
| ImmPRESS™ HRP Anti-Mouse IgG | Per manufacturer's instructions | Vector Laboratories | Catalog #MP-7402 |
| **Dissociation** | | | |
| Papain Dissociation System | Per manufacturer's instructions | Worthington Biochemical Corporation | Catalog #LK003150 |
| Hank's Balanced Salt Solution (HBSS) | 1× | Gibco | Catalog #14175-095 |
| D-(+)-Glucose | 6 g/L | Millipore Sigma | Catalog #G7021 |
| **Flow cytometry** | | | |
| FIX & PERM Cell Fixation & Cell Permeabilization Kit | Per manufacturer's instructions | Thermo Fisher Scientific | Catalog #GAS003 |
| Heat-inactivated Fetal Bovine Serum (HI-FBS) | 5% in FACS wash buffer | Gibco | Catalog #10437028 |

**Table a** (continued)

| Reagent | Concentration | Source | Identifier |
|---|---|---|---|
| Sodium azide (NaN₃) | 0.1% in FACS wash buffer | Fisher Scientific | Catalog #S2271-25 |
| PFA | 0.1% in sheath fluid | | |
| FxCycle Violet | 1:100 | Thermo Fisher Scientific | Catalog #F10347 |
| phospho-RB Alexa Fluor 488 | 1:50 | Cell Signaling Technology | Catalog #4277 |
| MyoD Allophycocyanin (APC) | 1:50 | Novus | Catalog #NBP2-34772APC |
| GFAP Alexa Fluor 647 | 1:50 | Cell Signaling Technology | Catalog #3657 |
| **Commercial assays** | | | |
| Drop-seq beads[32] | 90–110/μL | ChemGenes | Lot # 090316 |
| Nextera XT | NA | Illumina | FC-131-1024 |
| **Oligonucleotides** | | | |
| Template_Switch_Oligo: AAGCAG TGGTATCAACGCAGAGTGAA TrGrGrG | | 32 | N/A |
| TSO_PCR: AAGCAGTGGTATC AACGCAGAGT | | 32 | N/A |
| P5-TSO_Hybrid: AATGATAC GGCGACCACCGAGATCTACACG CCTGTCCGCGGAAGCAGTGGT ATCAACGCAGAGT*A*C | | 32 | N/A |
| Read1CustomSeqB: GCCTGTCCGCGGAAGCAG TGGTATCAACGCAGAGTAC | | 32 | N/A |

**Mice**. C57BL/6 mice, *SmoM2-eYFP*<sup>loxP/loxP</sup> mice and *Pham*<sup>loxP/loxP</sup> mice were purchased from the Jackson Laboratory. *Math1-Cre* mice, which express Cre recombinase under control of a cloned *Atoh1* promoter sequence[30], were generously shared by Dr. Robert Wechsler-Reya. *Gli-luc* mice, which express luciferase through GLI activation were generously shared by Dr. Oren Becher and Dr. Eric Holland. Mouse genotyping was performed using *Cre*, *Pham* or *SmoM2* primers above. All mice were of the species *Mus musculus* and maintained on a C57BL/6 background over at least five generations and were handled in compliance with all relevant ethical regulations for animal testing and research as specified by the University of North Carolina Institutional Animal Care and Use Committee in approved protocol 16-099.

**Pharmacological administration**. *M-Smo* littermates were injected intraperitoneally (I.P.) with 50 μL of 75 mg/kg vismodegib dissolved in one part of *N*-methyl-2-pyrrolidone (NMP) and nine parts of polyethylene glycol (PEG) or a vehicle control of 1:10 NMP:PEG. Equal numbers of female and male mice were randomly assigned to vehicle or vismodegib-treated groups. Mice were injected once daily for 3 days from P12 to P14 and collected at P15 for Drop-Seq, IHC and/or flow cytometry. For survival studies, mice were administered 50 μL of vehicle or 100 mg/kg vismodegib in NMP:PEG (1:10) once daily from P12 to P14 and then every other day. For long-term vismodegib studies, *M-Smo* mice underwent a similar protocol and were administered 50 μL of vehicle or 75 mg/kg vismodegib in NMP:PEG (1:10) once daily from P12 to P14 and then every other day and collected at P25. Mice were monitored daily for changes in body weight and symptoms such as ataxia, tremor and seizures. Animals were euthanized and brains collected if body weight dropped more than 20% over 24 h and/or they developed severe neurological symptoms according to approved protocols.

**Tissue preparation for Drop-Seq and flow cytometry**. Mice were anesthetized with isoflurane and euthanized by decapitation. Brains were cut in half sagittally and drop-fixed in 4% paraformaldehyde for IHC, and half cerebellum or tumor collected for Drop-Seq. Alternatively, half of the tumor was collected for flow cytometry.

Cells were dissociated using the Papain Dissociation System (Worthington Biochemical) according to the manufacturer's protocol, as in our prior studies[50]. Briefly, tumors or cerebella were dissected from isoflurane-anesthetized C57BL/6 or *M-Smo* mice and incubated in papain at 37 °C for 15 min. The tissue was then triturated, and the cells were spun down, resuspended and a density gradient was formed with ovomucoid inhibitor. Lastly, cells were resuspended in HBSS with 6 g/L glucose and diluted to ~100 cells/μl for Drop-seq.

Alternatively, cells were resuspended in Hank's Balanced Salt Solution with 6 g/L glucose, fixed and permeabilized for flow cytometry using the Fix & Perm Cell Fixation & Cell Permeabilization Kit (ThermoFisher Scientific), according to the manufacturer's protocol. Cells were stained for FxCycle Violet and with GFAP, phospho-RB, and/or MYOD1 as indicated. Samples were run on the Becton Dickinson LSR Fortressa and data were analyzed with FlowJo V10. The gating strategy for excluding debris, doublets and sub-G1 cells is exemplified in Supplementary Fig. 11.

**Immunofluorescence imaging**. Brains were fixed in 4% paraformaldehyde for at least 48 h. Tissue was processed and embedded in paraffin at the UNC Center for Gastrointestinal Biology and Disease Histology core. Sections were deparaffinized and antigen retrieval was performed using a low-pH citric acid-based buffer.

Staining was performed with assistance from the Translational Pathology Laboratories. Slides were scanned using the Leica Biosystems Aperio ImageScope software (12.3.3).

**IR-MALDESI**. Spatial distribution of vismodegib drug levels was studied by IR-MALDESI[51,52]. Sagittal sections (10 μm) of mouse brain samples were prepared in a cryotome, thaw-mounted on glass microscope slides, and maintained at −10 °C on the sample stage of the IR-MALDESI source chamber prior to analysis. The stage translated the sample step-wise across the focused beam of an IR laser ($\lambda$ = 2.94 μm, IR-Opolette 2371; Opotek, Carlsbad, CA, USA), which desorbed sample material from adjacent 100-μm-diameter sampling locations. An electrospray (50/50 mixture of methanol/water (v/v) with 0.2% formic acid) ionized the desorbed neutral molecules, and resulting ions were sampled into a high resolving power Thermo Fisher Scientific Q Exactive Plus (Bremen, Germany) mass spectrometer for synchronized analysis. The mass spectrometer was operated in positive ion mode from $m/z$ 200 to 800, with resolving power of $140{,}000_{FWHM}$ at $m/z$ 200. With high mass measurement accuracy (MMA) within 5 ppm maintained using protonated and sodiated adducts of diisooctyl phthalate as two internal lock masses at $m/z$ 391.28428 and 413.26623, vismodegib was identified by its protonated molecular ion $[M+H^+]^+$ at $m/z$ 421.01695. To generate images from mass spectrometry data, raw data from each voxel was converted to the mzXML format using MSConvert software[53]. These mzXML files were interrogated using MSiReader, a free software developed for processing MSI data[54].

**Drop-Seq library preparation and sequencing**. Drop-Seq libraries were prepared according to the Drop-seq protocol V3.1 (ref. [32]), with full details available online (http://mccarrolllab.com/dropseq/). Cell and bead concentrations were both set to between 95 and 110/μL.

WT cerebellum cells were co-encapsulated with barcoded beads using FlowJEM brand PDMS devices. Flow rates on the PDMS device for cells and beads were set to 3800 μL/h, flow rate for oil was maintained at 15,000 μL/h, resulting in a 4.5% bead occupancy rate in a 0.7-nL droplet.

Medulloblastoma cells were co-encapsulated using a Dolomite-brand glass device. All cells were processed within 1 h of tissue dissociation. Flow rates on the glass device were set to 2400 and 12,000 μL/h for cells/beads and oil, respectively, with a 1–2.5% bead occupancy rate.

Droplet breakage and library preparation steps followed Drop-seq protocol V3.1 (ref. [32]), with specific modifications:

- Following each PCR, an additional Ampure XP cleanup was performed at a 1× ratio, for a total of one 0.3× purification followed by a 1× purification. We found this to reduce residual PCR primer in the bioanalyzer electropherogram.
- Beads were stored at 4° after exonuclease step for up to 2 months prior to generating cDNA.

Following the completion of each set of experiments, a library pool consisting of the tagmented cDNA from 2000 beads/run was prepared and sequenced to low depth (~2.5 M reads/2K beads). These data were used to assess library efficiency, including total read losses to PolyA regions, nonsense barcodes and adapter sequences as well as the quality and number of the transcriptomes captured. Passable runs contained 40–60% of reads associated with the top 80–100 barcodes found in 2000 beads. For those runs that passed our quality assessment, we re-sequenced newly prepared libraries from the stored beads. The bulk cDNA libraries were prepared using the same ratio of 2000 beads/PCR.

**Processing of Drop-seq sequencing data**. Raw sequence data were processed in a Linux environment using Drop-seq Tools V1.13. (https://github.com/broadinstitute/Drop-seq/releases) to generate a digital expression (DGE) matrix. Step-by-step protocols may be found in the original documentation.

DGE matrices were used to generate Seurat objects in R (https://satijalab.org/seurat/). Input data are raw sequences in Fastq format, demultiplexed by sample identity. We first convert Fastq to BAM/SAM format and merge samples that were sequenced across multiple lanes.

The Drop-Seq Alignment pipeline version 1.13 (ref. [32]) was run following default settings, with the STAR aligner[55]. STAR version 2.5.4a was used to align against either an mm10, or an mm10 & hg19 mixed reference:

For Medulloblastoma Drop-Seq runs, cells were multiplexed with a spike-in of cultured HEK 293 cells to serve as an internal control. First, data were aligned against a mixed-species reference consisting of both hg19 and mm10 reference genomes, with chromosome and gene IDs annotated to contain a species-specific string. Cell barcodes were selected based on their human/mouse transcriptomic content, and only barcodes with >90% mouse transcript were chosen for further analysis.

Cells identified as mouse in origin were then aligned a second time, using a reference genome consisting of mm10, plus two synthetic chromosomes consisting of the Cre recombinase and SmoM2/EYFP fusion transcript transgenes.

WT cerebella were processed with no species-mixed spike-in and were only aligned once, against the mm10 transgenic reference.

**Filtering of Drop-seq sequencing data**. Full data analysis workflow and r scripts are available at https://github.com/ben-babcock/Gershon_single-cell.

Data analysis was performed in an R environment using the Seurat toolkit[37].

Following Seurat standard recommendations, data were first filtered for quality. Genes were required to be detected in as many as 30 cells to qualify as a "true" transcript—this is intended to prevent misaligned reads appearing as rare transcripts in the data. Cells were then filtered using specific QC criteria to limit the analysis to cells with transcriptomes that were well-characterized and not apoptotic.

We used filtering to identify which putative cells, identified by barcodes, represent informative cells. Putative cells with fewer than 500 UMIs or genes were considered to have too little information to be useful, and potentially to contain mostly ambient mRNA reads. Putative cells with greater than >4–5 standard deviations above the median nUMI or nGene were suspected to be doublets, improperly merged barcodes, or sequencing artifacts and were excluded. Putative cells with predominantly mitochondrial transcripts (>4–5 standard deviations above the median level of mitochondrial transcripts) were suspected to be dying cells and also excluded. For each QC step based on standard deviation, a cutoff between 4 and 5 standard deviations was selected in order to sample optimally around the mean, as visualized by violin plot following Macosko et al. guidelines[32]. Based on these considerations, QC criteria were:

- UMIs >500 and <4–5 standard deviations above the median
- Genes >500 <4–5 standard deviations above the median
- Mitochondrial transcripts <4–5 standard deviations above the median

In total, 84% of putative cells from *M-Smo* mice and 70% of putative cells from WT mice met QC criteria and were included in the analysis. From the 10 M-Smo mice, including both vehicle-treated and vismodegib-treated animals, we included a total of 29,234 cells, with five replicate mice per condition, with a range of 642–8062 cells per animal and a median of 2636 cells. From the five P7 WT mice, we included a total of 7090 cells, with a range of 788–2049 cells per animal and a median of 1169 cells.

**Clustering**. Genes were selected for differential expression across the sample using Seurat's highly variable gene selection tool, "FindVariableGenes". Mean expression and variance was calculated across the sample, and mean expression plotted against dispersion (variance/mean). Genes were sorted into equal-width bins and z-scored. We applied low and high mean expression cutoffs of 0.125 and 3 (x-cutoff) and minimum z-score of 0.5.

PCA was used to reduce the dimensionality of the gene expression matrix. A singular value decomposition (SVD) PCA was performed on the subset of highly variable genes. To identify an appropriate number of PCs, we employed a z-scoring method. We run a complete PCA reduction, and z-score the contribution of each PC to the total variance. PCs with $z > 2$ were considered significant and used further in analysis. The SVD PCA returns right singular values, representing the embeddings of each cell in PC space, and left singular values, representing the loadings (weights) of each genes in the PC space. Cell embeddings (right singular values) were weighted by the variance of each PC.

**Plotting for visualization**. Reduction to two dimensions was applied to the PCA matrix (the matrix of cells by their PC embeddings, hereafter referred to as PC-space) in order to place cells in a 2d plot for easy visualization (https://github.com/jkrijthe/Rt-SNE)[56]. In parallel, independent of the t-SNE projection, Louvain–Jaccard clustering was performed on the PC-space. This "bottom-up" clustering method employs a stochastic shared-nearest-neighbor (SNN) approach, in which cells are grouped according to their neighbors in PC space. The nearness of two cells is weighted by the Jaccard index, or the degree of sharing between the lists of each cell's nearest neighbors. The algorithm will build small groups of cells and attempt to iteratively merge them into clusters, until the modularity is max-imized. We found that a resolution of 0.8 was most appropriate for building biologically meaningful clusters.

**Cell-type identification**. Following PCA and t-SNE, we inspected clusters for expression of indicated markers. We determined differentially expressed genes using the Seurat implementation of the likelihood-ratio test for single cell gene expression[57]. Marker genes were plotted using an expression cutoff to facilitate the visualization of both high- and low-expression genes on a single plot. Cutoffs are applied so that only cells with expression >cutoff received the color corresponding to that gene. These cutoffs (gene expression thresholds) are found in Supplementary Dataset 9. In feature plots of multiple genes, for individual cells expressing multiple markers, each gene was over-plotted in the order presented in figure legends. To provide open access to our data through a convenient graphical user interface, we have set up an interactive data viewer accessible through the Gershon Lab web site (gershon-lab.med.unc.edu/single-cell).

**Doublet cell removal**. Doublets form when two cells' transcriptomes are co-captured on the same barcoded bead, resulting in two cells represented by a single barcode. Many doublets were removed during the initial filtering steps. However, we found clusters (or parts of clusters) that were identified by differential gene

expression to be doublets. The characteristic signature of these groups in our data is the significant differential expression of many genes with a low fold-change and high percent of cells expressing the gene in each group. These doublets were considered a technical effect and removed from further analysis of the sample.

**Merging clusters into nodes**. To classify clusters according to their similarities, we applied Seurat's "BuildClusterTree" function running on default parameters. This applies a hierarchical clustering algorithm to the cluster centroids in PC-space. Clusters closely apposed in the resulting dendrogram are referred to in the text as Nodes, owing to the fact that they are composed of clusters co-localized below a branch point, or node.

**Projecting tumor cells into WT t-SNE**. To place medulloblastoma cell types in the context of developmental biology, we performed PCA and cluster analysis on single-cell gene expression data from WT cerebellum harvested at P7. The PCA matrices for cells and genes were both extracted from the WT data.

Here the right singular values of the PCA represent a weighting (called Gene Loading) of each gene in WT PC-space. The left singular values (called Cell Embeddings) can be approximated for a given cell by multiplying the right singular values (gene weights) by a vector of the cells' gene expression. The sum of this product returns the left singular value.

We apply this method to approximate the PC-space embeddings for a cell from outside of the original PCA, here a medulloblastoma cell into WT PC-space.

For visualization purposes, we place each medulloblastoma (MB) cell into the WT t-SNE coordinate system using the k-NN algorithm. For each medulloblastoma cell, we identify the five most similar WT cells ($K = 5$). Considering these WT cells to be the five nearest neighbors, we average the t-SNE coordinates of these five WT cells to generate a t-SNE position for the projected MB cell. Plotting the projected cells overlaid on the WT cells demonstrates visually which WT and tumors cells were most similar and places the medulloblastoma cells in the biological context of a P7 cerebellum.

**ICA analysis**. To further understand the underlying biology of the medulloblastomas, we removed the stromal cells and performed an ICA. ICA was run on an input matrix of variable genes × cells expression data. To predict the number of significant IC dimensions, we perform a PCA on the same data, predict the number of significant PC dimensions as in the clustering steps, and proceed with an equivalent number of ICs. A t-SNE reduction of the ICs places cells into a two-dimensional space for visualization.

**Gene–gene correlation**. In Drop-Seq, as well as other single-cell transcriptomic methods, much of the expression data suffer from undersampling, or gene drop-out. To recover gene correlations across cells with missing values, we employed the MAGIC denoising method[43]. MAGIC shares information across similar cells to impute missing values by diffusing gene expression values across the neighbors. We found 12 neighbors to be the appropriate number which recovered statistical significance of biologically meaningful correlations.

**Reporting summary**. Further information on research design is available in the Nature Research Reporting Summary linked to this article.

## Data availability
The RNA sequencing data have been deposited in the Gene Expression Omnibus database under the accession code GSE129730. These data may be interactively explored at http://gershon-lab.med.unc.edu/single-cell/. Other datasets referenced during the study are available from the European Nucleotide Archive (https://www.ebi.ac.uk/ena/data/search?query=PRJEB23051) and the European Genome-phenome Archive (https://www.ebi.ac.uk/ega/studies/). All the other data supporting the findings of this study are available within the article and its supplementary information files and from the corresponding author upon reasonable request. A reporting summary for this article is available as a Supplementary Information file.

## Code availability
Scripts corresponding to the analysis contained in the current study are provided at https://github.com/ben-babcock/Gershon_single-cell.

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

## Acknowledgements

We thank the UNC CGBID Histology Core supported by P30 DK 034987, the UNC Tissue Pathology Laboratory Core supported by NCI CA016086 and UNC UCRF, and the UNC Neuroscience Center Confocal and Multiphoton Imaging and bioinformatics cores supported by The Eunice Kennedy Shriver National Institute of Child Health and Human Development (U54HD079124) and NINDS (P30NS045892). J.O. was supported by NINDS (F31NS100489). T.R.G. was supported by NINDS (R01NS088219, R01NS102627, R01NS106227) and by the UNC Department of Neurology Research Fund. T.R.G., K.W. and B.B. were supported by a TTSA grant from the NCTRACS Institute, which is supported by the National Center for Advancing Translational Sciences (NCATS), National Institutes of Health, through Grant Award Number UL1TR002489.

## Author contributions

J.K.O., B.B., D.M., A.C., K.W. and T.R.G. wrote the manuscript. J.K.O., B.B., D.M. and S.J.W. conducted most of the experiments and analysis of the data. L.L., J.M.S., M.J.Z., D.H., T.D., M.S., E.P.R., R.V., J.Z., O.S., M.V., I.E.-H., L.D.S., M.D.T., K.S.S. and P.A.N. generated data, performed analysis, or both. K.W. and T.R.G. were responsible for conception and oversight of the project. All authors discussed the results and reviewed the manuscript.

## Competing interests

The authors declare no competing interests.
