## [Peer Review File · Nature Communications]

Reviewers' comments:

Reviewer #1 (Remarks to the Author):

The manuscript entitled "Single cell RNA-seq shows cellular heterogeneity and lineage expansion in a mouse model of SHH-driven medulloblastoma support resistance to SHH inhibitor therapy" by Ocasio et al. examines intra-tumoral cell lineage diversity and response to treatment (SHH-inhibitor vismodegib) in a mouse model of SHH-MB (SmoM2 driven) using single-cell RNA-sequencing (scRNA-seq). The authors construct a picture of the cell types present within SHH-MB tumors, and present their relationship to the normally developing cerebellum. They show how these tumors contain a heterogeneous population of tumor cells that follow common cerebellar lineage programs (CGNP) and populations that express glial and stem cell markers. Further studies are performed on SHH-MBs treated with vismodegib, which shows a strong, but transient effect on SHH-MB growth. Utilizing the power of scRNA-seq, they show differential sensitivity of specific cellular clusters (Hes1 vs. MyoD1) to vismodegib, perform additional analysis on Sox2+ stem-like cells, and speculate on the role of polycomb components in the resistance of cells to vismodegib.

Overall the paper is well thought out, and provides strong data to support their major claims. The data is novel (this will be among the first MB scRNAseq manuscripts) and will be of interest to both basic researchers and clinicians due to their incorporation of vismodegib treatment resistance. While a degree of cellular heterogeneity has been previously described in SHH-MB tumors, the data presented both supports prior findings, and reveals additional heterogeneity within these groups based on expression of different glial and stem cell markers. The data on expansion of glial lineage following SmoM2 driven SHH signaling is also very interesting. The concern of intratumoral drug distribution was addressed with MALDESI.

My only minor concern is with the single time point of the drug resistant samples, but additional scRNA-seq experiments may be outside the scope of this study. Some of this may be able to be addressed with simple IF staining.

Minor concerns:

1) Does the sensitivity or resistance to vismodegib shape the overall composition of SHH-MB tumors at later time points? All scRNA-seq studies were performed on samples directly after 3 days of treatment, but later time points (x2 weeks) show increased pRB staining in tumors, and the Gli1 reporter data shows a bounce back to vehicle levels (which I would assume goes out past 3 day). To extrapolate from their data, one would expect that SHH-MB tumors chronically treated with vismodegib (and resistant), would be made of resistant cell types (MyoD1+), and contain few sensitive (Hes1+) cells. It would be interesting to look at this in vehicle and treated tumor collected at time of sacrifice due to symptoms. Do the Hes1+ cells rebound, or do they stay sensitive and remain lower? Again, while scRNA-seq would be beyond the scope of the study, the authors show nice IF staining with these markers, which could be used to give a baseline idea of these cell populations at late time points.

Typos:

Figure 1f is not listed in the figure legend text

Page 29 near the bottom (Additionally – spelling)

Page 31 near the bottom (inhibition – spelling)

Reviewer #2 (Remarks to the Author):

In the present manuscript the authors investigate the cellular heterogeneity and lineage commitment with and without treatment with the SMO inhibitor vismodegib in a mouse model of medulloblastoma using single cell RNAseq. This is an interesting topic and data obtained may shed

light on potential resistance mechanisms. The observation of cell populations at different stages of differentiation and with differential proliferation characteristics and response to vismodegib is expected. Overall the experimental design and data analysis are sound and the results are presented in a clear manner. However, the experimental model presents limitations with regard to the very rapid tumor development, continuous presence of active Cre and a polyclonal tumor development. Moreover, only a single dose of vismodegib was used and it is not clear how effective the inhibition of Hh-signaling was.

Specific comments;

- An improved documentation of the initial tumor regression would be useful.
- To further support the conclusion that Hh-pathway activation induce reacquired stem cell potential further validation is needed, e.g. by employing time resolved lineage tracing and/or additional Cre alleles.
- To support the conclusion about increased apoptosis upon treatment with vismodegib data should be presented.
- The proposed role of the PRC2-complex is not supported by any functional data and remains speculative.
- It is not clear if MyoD1+ cells showing apparent resistance to treatment with vismodegib are able to repopulate tumors and act as cancer stem cells.
- Could the inverted correlation between Gli1 and Sufu be due to the enrichment of differentiated cells in treated tumors? An analysis of this correlation involving Nodes A and B mapping back to differentiated and proliferative compartments may help to clarify this issue.
- Making the whole analysis pipeline available to the scientific community would be helpful.
- For the tSNE plots it should be described when a cell is considered positive for a particular gene and how collisions are handled.
- To validate the kNN analysis the use of a classifier is suggested or using the method recently suggested by Kobak et al. (<https://www.biorxiv.org/content/10.1101/453449v1>).
- Figure legend for fig 1f is missing. Number of replicates?
- For figure 4b it is suggested to specify which gene sets the ICs represent and to explain how they were defined. In figure 4e is the average expression defined by all cells or cells expressing at least one copy?
- Additional experiments are suggested to exclude that sorted Yfp+ cells are false positives.
- Statistical analysis of the data presented in fig 5 e-f should be included especially given the low number of cells.
- Are the data in fig 6 g,h fulfilling the independence assumption of the t-test?
- With regard to fig 7 quantitative analysis of single-cell data describing the number of Hes1+ and MyoD1+ cells would be interesting. In fig 7g the gates appear slightly shifted and a depletion is apparent also for MyoD1. Quantitative data would help to resolve this.

Reviewer #3 (Remarks to the Author):

In this article, Ocasio et al. present single-cell transcriptomic analyses of a mouse model of the SHH subtype of medulloblastoma with and without treatment with the SMO inhibitor Vismodegib. They identify cellular heterogeneity within the tumors, as well as some correlative features of cell types and genes expressed in populations in each condition. There is clearly a need to understand the underlying cellular heterogeneity of medulloblastoma and how this influences treatment response. The main challenge with the manuscript is that it lacks focus and includes initial observations that do not have sufficient support to justify the conclusions put forth by the authors. The manuscript would be significantly strengthened if the authors focused on one pathway and provided convincing new insight into the mechanisms of Vismodegib resistance and/or new insight into how to combine other targeted therapies with Vismodegib.

More specifically, my major concerns are:

- 1) The way the authors present the data makes the story difficult to follow. For example, why do they present data in figure 2 using both tSNE and ICA? How did they select 3 clusters with ICA that they then use going forward? Which cells within the tSNE are tumor cells, can the authors use YFP or another marker to help orient the reader?
- 2) How do the authors project vehicle-treated cells onto the ICA in figure 3? Shouldn't the tumor and normal cells be clustered together to see the similarity between the two rather than projecting one cell type onto the other? If so, I would expect we would see the tumor cells cluster separately from normal cells as there is aberrant expression of genes in the tumor cells. Where are all the other normal cell types, such as gluta/gaba DCN, UBCs, and purkinje cells? Using P7 mice also excludes progenitor populations that would be seen in embryonic cerebella. Without showing all those cell types, it is not convincing that the tumor cells only resemble granule cells.
- 3) Figure 4 does not add a great deal and could probably be combined with figure 3. Have the authors done analyses that control for cell cycle to see if there are distinct cell types that are being obscured by the cell cycle genes?
- 4) . The purpose of figure 5 is difficult to follow. They may see different groups of cells within the glial lineages, but that is not followed up later in the paper with experiments that support specific conclusions (such as there is a new cell type).
- 5) For figure 6, the comparison of treated/control cells only need to be done once. It is not clear what the extra panels add. Panels g and h contain the important new information.
- 6) Figure 7-8 present preliminary results that could be used as confirmatory, but do not support conclusion about new insight into Vismodegib resistance. Picking one new finding and providing multiple lines of evidence would be more impactful. It is also unclear what conclusions the authors are drawing from the bulk RNA-seq. Isn't the point that there is heterogeneous expression of those genes in the tumor and that they are present in all tumors?

Minor Concerns:

- 1) What are defined as "neurons"? Are they a specific type of neuron found in the cerebellum?
- 2) The authors speculate that the YFP expressing glial cells are tumor cells. It is possible there is just transient, aberrant Atoh1 expression in those cells.

We appreciate the in-depth reviews and we have addressed each of the reviewer concerns as described below. We thank the reviewers for the feedback, which has helped us to strengthen the manuscript.

Point by point response to the Reviewers' comments:

Reviewer #1 (Remarks to the Author):

The manuscript entitled “Single cell RNA-seq shows cellular heterogeneity and lineage expansion in a mouse model of SHH-driven medulloblastoma support resistance to SHH inhibitor therapy” by Ocasio et al. examines intra-tumoral cell lineage diversity and response to treatment (SHH-inhibitor vismodegib) in a mouse model of SHH-MB (SmoM2 driven) using single-cell RNA-sequencing (scRNA-seq). The authors construct a picture of the cell types present within SHH-MB tumors, and present their relationship to the normally developing cerebellum. They show how these tumors contain a heterogeneous population of tumor cells that follow common cerebellar lineage programs (CGNP) and populations that express glial and stem cell markers. Further studies are performed on SHH-MBs treated with vismodegib, which shows a strong, but transient effect on SHH-MB growth. Utilizing the power of scRNA-seq, they show differential sensitivity of specific cellular clusters (Hes1 vs. MyoD1) to vismodegib, perform additional analysis on Sox2+ stem-like cells, and speculate on the role of polycomb components in the resistance of cells to vismodegib.

Overall the paper is well thought out, and provides strong data to support their major claims. The data is novel (this will be among the first MB scRNAseq manuscripts) and will be of interest to both basic researchers and clinicians due to their incorporation of vismodegib treatment resistance. While a degree of cellular heterogeneity has been previously described in SHH-MB tumors, the data presented both supports prior findings, and reveals additional heterogeneity within these groups based on expression of different glial and stem cell markers. The data on expansion of glial lineage following SmoM2 driven SHH signaling is also very interesting. The concern of intratumoral drug distribution was addressed with MALDESI.

My only minor concern is with the single time point of the drug resistant samples, but additional scRNA-seq experiments may be outside the scope of this study. Some of this may be able to be addressed with simple IF staining.

We thank the Reviewer for the overall positive feedback and for the helpful suggestions below.

Minor concerns:

1) Does the sensitivity or resistance to vismodegib shape the overall composition of SHH-MB tumors at later time points? All scRNA-seq studies were performed on samples directly after 3 days of treatment, but later time points (x2 weeks) show increased pRB staining in tumors, and the Gli1 reporter data shows a bounce back to vehicle levels (which I would assume goes out past 3 day). To extrapolate from their

data, one would expect that SHH-MB tumors chronically treated with vismodegib (and resistant), would be made of resistant cell types (MyoD1+), and contain few sensitive (Hes1+) cells. It would be interesting to look at this in vehicle and treated tumor collected at time of sacrifice due to symptoms. Do the Hes1+ cells rebound, or do they stay sensitive and remain lower? Again, while scRNA-seq would be beyond the scope of the study, the authors show nice IF staining with these markers, which could be used to give a baseline idea of these cell populations at late time points.

We have added new experiments, in which medulloblastoma-bearing *M-Smo* mice were treated with vismodegib for 14 days, starting at P12 and then the brains were harvested and the tumors were stained for HES1 and pRB or Myod1 and pRB. These data show that the highly proliferative HES1⁺ population of tumor cells that expressed both HES1 and pRB remained suppressed. In contrast, MYOD1+ cells that were also pRB⁺ remained numerous. Thus, the sensitivity and resistance to vismodegib continued to shape the overall composition of the tumor at later time points. These data are presented in Supplemental Figure 6.

Typos:

Figure 1f is not listed in the figure legend text

We have corrected the figure legend and revised the figure to include statistical analysis and to make the number of replicates apparent.

Page 29 near the bottom (Additionally – spelling)

Page 31 near the bottom (inhibition – spelling)

We have corrected these errors

Reviewer #2 (Remarks to the Author):

In the present manuscript the authors investigate the cellular heterogeneity and lineage commitment with and without treatment with the SMO inhibitor vismodegib in a mouse model of medulloblastoma using single cell RNAseq. This is an interesting topic and data obtained may shed light on potential resistance mechanisms. The observation of cell populations at different stages of differentiation and with differential proliferation characteristics and response to vismodegib is expected. Overall the experimental design and data analysis are sound and the results are presented in a clear manner.

We appreciate the finding that the design and data analysis are sound and the presentation is clear.

However, the experimental model presents limitations with regard to the very rapid tumor development, continuous presence of active Cre and a polyclonal tumor development. Moreover, only a single dose of vismodegib was used and it is not clear how effective the inhibition of Hh-signaling was.

We have added additional quantitative studies of pRB suppression and statistical analysis and Gli-luc signal to further demonstrate SHH suppression. We selected the vismodegib dose based on the MTD, so as to deliver highest level of SHH inhibition within the therapeutic window. We agree that at this dose, there was a heterogeneous suppression of SHH signaling. However, we do not agree that the extent of suppression is unclear. The effectiveness of SHH inhibition was demonstrated by the suppression of SHH activation markers *Gli1*, *Ptch1*, *Hhip* and *Sfrp1* in the vismodegib-responsive cells. Moreover, the MALDESI data show even distribution of the drug in the tumor. Together, these data show vismodegib penetrated all regions of the tumors and suppressed SHH signaling in the cells that were responsive. The persistence of SHH activation in the resistant cells is a central finding of our work.

Specific comments;

-- An improved documentation of the initial tumor regression would be useful.

We have added a quantitative analysis of pRB expression, comparing treated and untreated tumors after 3 days, and vismodegib-treated tumors after 2 weeks on therapy. These data, added to Figure 1, effectively demonstrate that tumors regressed after 3 days of treatment, and began to recur over 2 weeks of therapy.

-- To further support the conclusion that Hh-pathway activation induce reacquired stem cell potential further validation is needed, e.g. by employing time resolved lineage tracing and/or additional Cre alleles.

We do not conclude that Hh-pathway activation induces stem cell potential, but rather that Hh pathway activation expanded the potential of the Math1 lineage to generate glial phenotypes. As further support to this specific conclusion, we have added new experiments that use flow cytometry with the Cre-activated DENDRA2 marker bred into the medulloblastoma prone *M-Smo* mice. These new studies provide an alternative method for identifying Math1-Cre lineage tumor cells, in addition to *Yfp* expression. We use flow cytometry to quantify both DENDRA2 and GFAP expression. These data show that Math1-lineage cells generate GFAP+ progeny specifically in tumors, in sharp contrast to Math1-lineage cells in normal cerebella. The data have been added to the revised Figure 5.

We have removed data describing the expression of Nestin and vimentin in neural progenitor-like tumor cells in order to avoid conclusions about stem cell phenotype that are not supported by direct evidence.

-- To support the conclusion about increased apoptosis upon treatment with vismodegib data should be presented.

To address this issue, we performed cleaved caspase 3 studies, but these did not show increased apoptosis. As shown in the new Supplemental Fig 4, cleaved caspase 3 staining after vismodegib treatment did not show increased cell death. In the absence of direct evidence, we clarify the speculative nature of the conclusion that cell

death may be increased. We revised the text by discussing that absence of increased cleaved caspase 3 may reflect either no increase in cell death or a small, asynchronous increase in cell death. We added a comment that studies in Bax-mutant tumors will be needed to resolve definitively if apoptosis is increased, as suggested by the absence of accumulated neurons.

-- The proposed role of the PRC2-complex is not supported by any functional data and remains speculative.

We agree that this proposed role is speculative. Functional studies with PRC2 component inhibitors or mutant mice will be needed to evaluate this possibility, but go beyond the scope of the work. We have revised the text to state more clearly that the proposed role of the PRC2 complex is speculative.

-- It is not clear if MyoD1+ cells showing apparent resistance to treatment with vismodegib are able to repopulate tumors and act as cancer stem cells.

Our new data in mice treated for 14 days with vismodegib show that MyoD1+ cells remain pRB positive over the course of prolonged treatment. These data, in Supplemental Figure 6, support our description of MyoD1+ cells as a population that continues to proliferate. We do not state that these are cancer stem cells, but rather that MYOD1+ cells are one of the sub-populations that persist in a proliferative state and that may drive recurrence.

-- Could the inverted correlation between Gli1 and Sufu be due to the enrichment of differentiated cells in treated tumors? An analysis of this correlation involving Nodes A and B mapping back to differentiated and proliferative compartments may help to clarify this issue.

We agree that the treated tumors contained highly differentiated cell types that were not found in the controls. In order to compare across similar cell types, we added a Supplemental Figure 8, in which we removed the cells of Node D, which was not represented in controls. We further added a color code identifying the Node for each cell in the plot. This new figure makes clear that specifically in the treated tumors, cells with higher Sufu tend to group in Node C and to show relatively more differentiation, and cells with lower Sufu tend to group in Nodes A and B and to remain undifferentiated and proliferative. In contrast, in untreated tumors, higher Sufu varies directly with proliferation.

-- Making the whole analysis pipeline available to the scientific community would be helpful.

We have made the code available at: github.com/ben-babcock/Gershon_single-cell, and we have added this link to the Supplemental Methods. We have also added a link to a web-based application (gershon-lab.med.unc.edu/single-cell/) that allows readers to plot the expression of any gene across the t-SNEs for the WT P7 cerebellum

and combined treated and untreated tumors. The link to this application has been added to the first paragraph of the Discussion and to the Data Availability statement.

-- For the t-SNE plots it should be described when a cell is considered positive for a particular gene and how collisions are handled.

We have added Supplementary Table 9, which shows the threshold for each marker. Moreover, our web app allows readers to make new t-SNE plots with different marker thresholds.

Collisions, where individual cells express more than one of the plotted markers, occur for the first time in Figure 3. In the revised Figure 3 legend, we have added a description of how cells that are positive for multiple markers are shown in multicolor t-SNE plots. Additionally, the web app that we have added at gershon-lab.med.unc.edu/single-cell/ allows readers to review individual t-SNE plots to resolve the expression of individual genes shown in multicolor plots.

-- To validate the kNN analysis the use of a classifier is suggested or using the method recently suggested by Kobak et al. (<https://www.biorxiv.org/content/10.1101/453449v1>).

This method is not suitable for our analysis because our kNN was performed on PCs, rather than on genes. Because each PC contains a large amount of information, we cannot drop individual PCs and determine if the classification remains unchanged. As an alternative method of validation, we state in the text that the kNN method is validated by the accurate matching of each stromal cell type from tumor to corresponding clusters in the WT cerebellum.

-- Figure legend for fig 1f is missing. Number of replicates?

As stated in response to Reviewer 1, we have added the missing figure legend and included the number of replicates.

-- For figure 4b it is suggested to specify which gene sets the ICs represent and to explain how they were defined.

We adjusted the text to make clear that that gene sets are specified in Supplemental Table 3. Further, we concluded that there was no systematic way to associate each IC with a biologic process and therefore limited the characterization of ICs to IC2, which was unambiguously associated with neuronal differentiation.

In figure 4e is the average expression defined by all cells or cells expressing at least one copy?

We have added text to the legend clarifying that average expression is defined by expression in all cells, including cells where 0 transcripts are detected.

-- Additional experiments are suggested to exclude that sorted Yfp+ cells are false positives.

The new studies use DENDRA2 lineage tracing, detected by flow cytometry, as an alternative approach to show GFAP⁺ cells descended from Math1-Cre-expressing predecessors. The fraction of cells that were GFAP⁺ in the DENDRA2⁺ population closely matches the fraction of cells that were *Gfap*⁺ in the *Yfp*⁺ population, validating the detection of *Yfp* in glial cells.

-- Statistical analysis of the data presented in fig 5 e-f should be included especially given the low number of cells.

We added p values derived from Fisher's exact test to the data presented in 5e-f and reference these p values in the relevant section of Results. The new analysis shows that the *Yfp*⁺ astrocytic cells are significantly enriched in the undifferentiated subset, while the distribution of *Yfp*⁺ oligodendrocytic cells between differentiated and undifferentiated subsets is not statistically significant. These findings are described in detail in the revised Results.

-- Are the data in fig 6 g,h fulfilling the independence assumption of the t-test?

The t-test is appropriate since comparisons are between the number of cells in each cell type across treatment groups. For each individual cell type, the population size in each treatment group is independent.

-- With regard to fig 7 quantitative analysis of single-cell data describing the number of Hes1⁺ and MyoD1⁺ cells would be interesting.

We have added quantitative data in a new panel 7g, as suggested.

In fig 7g the gates appear slightly shifted and a depletion is apparent also for MyoD1. Quantitative data would help to resolve this.

We have corrected the gates to be the same for all replicates as now shown in both graphs of the new 7h. We have also added the mean \pm SEM for the MYOD1⁺ cells in both genotypes. We revised the Results to state that the data show that there is a trend toward a reduction of MYOD1⁺ cells, but that in contrast to the HES1⁺ cells, MYOD1⁺ cells remain proliferative after treatment.

Reviewer #3 (Remarks to the Author):

In this article, Ocasio et al. present single-cell transcriptomic analyses of a mouse model of the SHH subtype of medulloblastoma with and without treatment with the SMO inhibitor Vismodegib. They identify cellular heterogeneity within the tumors, as well as some correlative features of cell types and genes expressed in populations in each condition. There is clearly a need to understand the underlying cellular heterogeneity of medulloblastoma and how this influences treatment response.

We appreciate the comment that there is a need to understand how heterogeneity influences tumor treatment, which is a major concern of this work.

The main challenge with the manuscript is that it lacks focus and includes initial observations that do not have sufficient support to justify the conclusions put forth by the authors. The manuscript would be significantly strengthened if the authors focused on one pathway and provided convincing new insight into the mechanisms of Vismodegib resistance and/or new insight into how to combine other targeted therapies with Vismodegib.

Our changes in response to feedback from all 3 reviewers have added clarity and thematic focus. We have also added new data to support the specific conclusion that pluripotency is increased within the tumors. In this new data, we use fluorescent lineage tracing to confirm by alternative methods that Math1-lineage cells in tumors take on glial fates outside of the expected fates of the Math1 lineage. For other conclusions that remain speculative, we acknowledge the need for more information and identify follow-up experiments that may prove or disprove our interpretations.

More specifically, my major concerns are:

1) The way the authors present the data makes the story difficult to follow. For example, why do they present data in figure 2 using both t-SNE and ICA? How did they select 3 clusters with ICA that they then use going forward?

As clarified in the revised text, we identified clusters using PCA, and we subsequently used hierarchical clustering analysis (HCA), for the specific purpose of developing “an ordered subclassification of the cells within the multi-cluster complex”. This HCA showed that the 9 clusters of the multi-cluster complex were more similar to one another than to the clusters identified as stromal cell types, and defined 3 groups which were designated as Nodes A-C.

can the authors use YFP or another marker to help orient the reader?

We considered adding *Yfp* to the results at this section in response to this comment. However, we found that clarity was not improved, because of the variety of cell types that express *Yfp*. We found that it is important first to discuss the similarity of each cluster to recognizable cell types, before discussing the *Yfp* lineage tracing. However, the new text states the purpose of the HCA more clearly, which may address the underlying issue.

2) How do the authors project vehicle-treated cells onto the ICA in figure 3? Shouldn't the tumor and normal cells be clustered together to see the similarity between the two rather than projecting one cell type onto the other? If so, I would expect we would see the tumor cells cluster separately from normal cells as there is aberrant expression of genes in the tumor cells.

We agree that in a cluster analysis of WT and tumor cells considered together, tumor cells and WT would group separately from one another, because of the differences between them. However, our purpose is here is to analyze the similarities rather than the differences. We revised the text to state that the k-NN “algorithm sorted

tumor cells according to their similarity to the cell types present in the WT dataset, and determined their best fit among these types”.

Where are all the other normal cell types, such as gluta/gaba DCN, UBCs, and purkinje cells?

To address this question, we analyzed the expression of glutamatergic marker *Slc17a6* (aka *vGlut2*) and gabaergic marker *Gad1* (aka *Gad67*) in our single cell data sets. We have added new feature plots as Supplemental Figure 1, which identify cells expressing *Slc17a6*, *Gad*, Purkinje cell marker *Calb1* and CGNP marker *Calb2*. These plots highlight the positions of identifiable populations on the WT P7 and tumor t-SNE projections. Readers can further identify cell types by plotting additional markers through our newly developed web-based application at <http://gershon-lab.med.unc.edu/single-cell/>.

Using P7 mice also excludes progenitor populations that would be seen in embryonic cerebella. Without showing all those cell types, it is not convincing that the tumor cells only resemble granule cells.

As described in the text, the P7 brain was selected because it contains cells in a range of differentiation states, providing a line-up of potential matches for tumor cells mapped by kNN to the WT t-SNE projection. We clarify our point in the revision, which is that within the P7 cerebellum, the subset of cells in Nodes A-C are more similar to the CGNPs than to other cell types in the WT P7 cerebellum, and that Nodes A-C correspond to progressively more differentiated types of CGNPs.

3) Figure 4 does not add a great deal and could probably be combined with figure 3. Have the authors done analyses that control for cell cycle to see if there are distinct cell types that are being obscured by the cell cycle genes?

We have revised the results section describing Figure 4 by simplifying the consideration of the ICA and revised Figure 4 by color mapping only the ICA corresponding to differentiation. The revised figure more clearly shows the alternate processes of differentiation and cell cycle re-entry.

At the reviewer’s suggestion, we examined whether regressing against genes known to mark cell cycle state would identify new cell types. We defined the set of cell cycle genes based on the previously published single cell paper that we cite (Macosko et al 2015). We transformed our data by scaling to eliminate the effect of enrichment for these cell cycle genes. This transformation affected the number of significant PCs (decrease to 9 from 10) and individual cell positions in the t-SNE, but no new cell types emerged. We did not add this analysis to the manuscript, since it did not alter the conclusions of the paper.

4) . The purpose of figure 5 is difficult to follow. They may see different groups of cells within the glial lineages, but that is not followed up later in the paper with experiments that support specific conclusions (such as there is a new cell type).

The purpose of the figure is to parse the overlap between lineage tracing, Sox2 expression, glial phenotype, and stem cell phenotype. We have revised the text describing Figure 5 to be more clear, and we revised the figure by adding new data.

The new Figure 5 now includes flow cytometry studies of Cre-activated DENDRA2 to trace the Math1 lineage in our tumor model. These data show that Math1-lineage cells give rise to GFAP+ cells specifically in tumors. The data support our conclusion, based on *Yfp*-lineage tracing in single cell data, that the Math1 lineage in tumors includes glia. The rest of the figure analyzes the overlapping distributions of Sox2 expression, *Yfp* expression and stem cell and differentiation markers and demonstrates that Math1 lineage cells give rise to Sox2+ different subsets with glial, neural progenitor and stem cell markers.

5) For figure 6, the comparison of treated/control cells only need to be done once. It is not clear what the extra panels add. Panels g and h contain the important new information.

We have revised Figure 6 to address this point. The HCA panel is needed to develop the classification into 4 Nodes, and the marker dot plot shows the emergence of differentiation across the nodes. We consider that the treated versus control comparisons of Node distribution, *Yfp* expression, P7-kNN projections and ICA to be important information that should remain in the figure. However, we agree that the panels showing the combined treated and controls do not need to be reiterated and we have removed them.

6) Figure 7-8 present preliminary results that could be used as confirmatory, but do not support conclusion about new insight into Vismodegib resistance. Picking one new finding and providing multiple lines of evidence would be more impactful.

Figure 7 presents data on the differential sensitivity of HES1+ and MYOD1+ cells that is not speculative. This data is central to our conclusion that different subsets of tumor cells have different responses to treatment.

Figure 8 includes data on the correlation of PRC2 complex genes and SUFU with treatment sensitivity. We agree that the idea of a causative link between expression of PRC2 genes, SUFU and resistance to vismodegib is speculative. We contend that this level of speculation late in the paper is reasonable and additional experiments to test for causality, such as deleting PRC2 genes in tumors and looking at the effect on vismodegib sensitivity are beyond the scope of the work. However, we added further analysis of SUFU as Supplemental Figure C.

It is also unclear what conclusions the authors are drawing from the bulk RNA-seq. Isn't the point that there is heterogeneous expression of those genes in the tumor and that they are present in all tumors?

The bulk RNA-seq studies show data from human tumors and are included to show the relevance of genes identified in the model to the actual cancer that occurs in patients.

Minor Concerns:

1) What are defined as "neurons"? Are they a specific type of neuron found in the cerebellum?

As the new feature plots of *Gad1*, *Slc16a6*, *Calb1* and *Calb2* in Supplemental Figure 1 make clear, the cells previously labelled "neurons" are CGNs. We have updated the label accordingly.

2) The authors speculate that the YFP expressing glial cells are tumor cells. It is possible there is just transient, aberrant *Atoh1* expression in those cells.

It is not clear what would distinguish cells with transient *Atoh1* (*Math1*) expression from cells of the *Math1* lineage. Any cells that express *Atoh1*, either transiently and durably, are in the *Math1* lineage, as are their progeny. Our point is that in medulloblastomas, the *Math1* lineage, defined by activation of *Math1-Cre*, includes glial cells, whereas in normal cerebellum the *Math1* lineage is restricted to neurons. Based on the *Math1* lineage and the unexpected expression of glial markers by *Math1* lineage cells, we consider cells expressing both *Yfp* AND *Gfap*, or DENDRA2 AND GFAP, to be tumor cells.

Reviewers' comments:

Reviewer #1 (Remarks to the Author):

The authors have addressed my one raised concern about the composition of the resistant population at a later time point by adding an additional 2 week time point in Supplemental figure 6. This helps support their findings at the shorter time (3 days), suggesting this resistance phenotype initially described continues to be relevant in disease progression. This data, along with the other revisions the authors have made, strengthen the overall narrative of the article. I would recommend it for publication.

Reviewer #4 (Remarks to the Author): recruited to replace Reviewer #2; expertise in HH signalling

Ocasio et al. have explored cellular populations responsive and resistant to a SHH inhibitor, Vismodegib within a murine mouse model using a single cell RNA-seq technology. The revised manuscript was too descriptive and lacks requested validations of Vismodegib-resistant populations and signaling pathways causing chemo-resistance, although some of questions were properly solved by the revised. Eventually, few novel mechanistic insights were shown by further biological analyses based on their single cell RNA-seq data. I would expect them to expand their work till functional analyses, which is one of advantages of murine models of cancers as requested in the first round of the reviews.

Here are my specific comments on a few of their responses to the reviewer #2:

-- To further support the conclusion that Hh-pathway activation induce reacquired stem cell potential further validation is needed, e.g. by employing time resolved lineage tracing and/or additional Cre alleles.

We do not conclude that Hh-pathway activation induces stem cell potential, but rather that Hh pathway activation expanded the potential of the Math1 lineage to generate glial phenotypes. As further support to this specific conclusion, we have added new experiments that use flow cytometry with the Cre-activated DENDRA2 marker bred into the medulloblastoma prone M-Smo mice. These new studies provide an alternative method for identifying Math1-Cre lineage tumor cells, in addition to Yfp expression. We use flow cytometry to quantify both DENDRA2 and GFAP expression. These data show that Math1-lineage cells generate GFAP+ progeny specifically in tumors, in sharp contrast to Math1-lineage cells in normal cerebella. The data have been added to the revised Figure 5. We have removed data describing the expression of Nestin and vimentin in neural progenitor-like tumor cells in order to avoid conclusions about stem cell phenotype that are not supported by direct evidence.

As also commented by authors, Atoh1-Cre mice does not guarantee that Cre is expressed only in Atoh1-expressing cells. Thus, the data still remains two possibilities: 1) Atoh1+ cells, such as granule cell progenitors differentiated glial cell progenies, followed by GFAP+ tumor cells or 2) Cre-expressing GFAP+ glial progenitor cells transformed GFAP+ tumor cells. Their data cannot tell both, thus the 5th section in the RESULT part has not given a rigid conclusion yet.

-- Making the whole analysis pipeline available to the scientific community would be helpful. We have made the code available at: github.com/ben-babcock/Gershon_single-cell, and we have added this link to the Supplemental Methods. We have also added a link to a web-based application (gershon-lab.med.unc.edu/single-cell/) that allows readers to plot the expression of any gene across the t-SNEs for the WT P7 cerebellum and combined treated and untreated tumors. The link to this application has been added to the first paragraph of the Discussion and to the Data Availability statement.

The link the authors indicated did not show the pipeline. It's only the visualization segments. They need to explain the details of the whole analysis pipeline.

Reviewer #5 (Remarks to the Author): Recruited to replace Reviewer #3; expertise in single cell sequencing

The manuscript "Single-cell RNA-seq shows cellular heterogeneity and lineage expansion in a mouse model of SHH-driven medulloblastoma support resistance to SHH inhibitor therapy" is a timely and well designed study. The team uses single cell RNA-seq in a novel and creative way and their conclusions will be of great interest to the readers.

However, I would like a few comments addressed before publication:

- For the initial vehicle-treated scRNAseq experiments in Figure 2: How many cells were analyzed total and per mouse? How many passed basic QC and how many failed? Based on what parameters (only >500 genes expressed is mentioned in methods) and where were the thresholds? Basic stats and transparency about these data are a must in single-cell analysis.
- Other than clustering and a rather sparse YFP expression, I am not confident about which cells are tumor cells and which are normal cells. Is there a way to call specific mutations or increase the sensitivity of YFP calling? Would these SHH-Mb have CNVs that could be inferred from the data and used as a way to distinguish normal from tumor cells?
- For the projection of the MB cells onto the normal cerebellum, please use a second method for validation, e.g Seurat CCA.
- The authors should also apply the mouse MB data to recently published mouse cerebellum data sets, e.g. Carter RA and al, Curr Biol 2018; Hovestadt V. et al, Nature 2019; Vladoiu MC et al, Nature 2019.
- I am not sure the last part about PRC2 has strong enough data to support the conclusion in the setting of low gene sensitivity/low coverage per cell. Please address or rather leave out of the paper.

Reviewers' comments:

Reviewer #1 (Remarks to the Author):

The authors have addressed my one raised concern about the composition of the resistant population at a later time point by adding an additional 2 week time point in Supplemental figure 6. This helps support their findings at the shorter time (3 days), suggesting this resistance phenotype initially described continues to be relevant in disease progression. This data, along with the other revisions the authors have made, strengthen the overall narrative of the article. I would recommend it for publication.

We appreciate the reviewer noting that we have addressed the cited concern, and that the additional changes have strengthened the work. We are grateful for the suggested experiment.

Reviewer #4 (Remarks to the Author): recruited to replace Reviewer #2; expertise in HH signalling

Ocasio et al. have explored cellular populations responsive and resistant to a SHH inhibitor, Vismodegib within a murine mouse model using a single cell RNA-seq technology. The revised manuscript was too descriptive...

We disagree with the characterization of our manuscript as "too descriptive". The central theme of the work is that it compares tumors under two different experimental conditions, untreated vs vismodegib-treated, analyzed by scRNA-seq. The use of scRNA-seq to study the effect of anti-tumor treatment on tumor heterogeneity is both experimental and innovative.

and lacks requested validations of Vismodegib-resistant populations...

We ask the reviewer to acknowledge that we added requested validations of vismodegib-resistant populations. Specifically, we added the analysis of tumor after 2 weeks of treatment, as requested in the prior round of review, for the purpose of validating the stability of vismodegib resistant populations. This addition was acknowledged by Reviewer 1 and found to satisfy the raised concerns.

and signaling pathways causing chemo-resistance,

We show that persistent SHH pathway activation is the cause of resistance to the SMO inhibitor. Resistant cells show continued expression of the SHH pathway transcription factor *Gli1* (Fig. 7a), *Ptch1*, *Hhip*, and *Sfrp1* (Supplementary Fig. 5). We acknowledge that our hypothesis that low SUFU expression contributes to persistent SHH activation in resistant cells is more speculative. We addressed reviewer concern about the complexity of analyzing *Sufu:Gli1* correlation by adding specific reviewer-

requested analysis of SUFU expression. We acknowledge that we did not add new validation of our speculation regarding a contribution of the PRC2 complex to resistance. To meet reviewer concern for validation, we have now removed the figures highlighting PRC2 complex genes in vismodegib-resistant cells, which we agree were inconclusive in their implication.

although some of questions were properly solved by the revised. Eventually, few novel mechanistic insights were shown by further biological analyses based on their single cell RNA-seq data.

Novel mechanistic insights from our data include our finding that tumors contain large populations of resistant cells as early as three days into treatment, and that resistance is mediated by SMO-independent SHH pathway activation, as demonstrated by the persistence of Gli1 and other SHH target genes in resistant cells. This information on the mechanism of targeted inhibitor failure is new and could not be obtained without single cell analysis.

I would expect them to expand their work till functional analyses, which is one of advantages of murine models of cancers as requested in the first round of the reviews.

Functional analyses, such as genetic deletion studies, were not requested in the first round of review. While we share the reviewer's interest in such studies, we do not agree that these studies should be required prior to the publication of our current data.

Here are my specific comments on a few of their responses to the reviewer #2:

-- (citing Reviewer 2's prior comment) To further support the conclusion that Hh-pathway activation induce reacquired stem cell potential further validation is needed, e.g. by employing time resolved lineage tracing and/or additional Cre alleles.

(Our response to this prior comment) We do not conclude that Hh-pathway activation induces stem cell potential, but rather that Hh pathway activation expanded the potential of the Math1 lineage to generate glial phenotypes. As further support to this specific conclusion, we have added new experiments that use flow cytometry with the Cre-activated DENDRA2 marker bred into the medulloblastoma prone M-Smo mice. These new studies provide an alternative method for identifying Math1-Cre lineage tumor cells, in addition to Yfp expression. We use flow cytometry to quantify both DENDRA2 and GFAP expression. These data show that Math1-lineage cells generate GFAP+ progeny specifically in tumors, in sharp contrast to Math1-lineage cells in normal cerebella. The data have been added to the revised Figure 5. We have removed data describing the expression of Nestin and vimentin in neural progenitor-like tumor cells in order to avoid conclusions about stem cell phenotype that are not supported by direct evidence.

As also commented by authors, Atoh1-Cre mice does not guarantee that Cre is expressed only in Atoh1-expressing cells. Thus, the data still remains two possibilities:

1) *Atoh1*+ cells, such as granule cell progenitors differentiated glial cell progenies, followed by GFAP+ tumor cells or 2) Cre-expressing GFAP+ glial progenitor cells transformed GFAP+ tumor cells. Their data cannot tell both, thus the 5th section in the RESULT part has not given a rigid conclusion yet.

The reviewer suggests 2 possible conclusions could be drawn from the data, and argues that we cannot distinguish between them. These suggested conclusions are: 1) that GFAP- cells of the *Math1* lineage become transformed by *SmoM2* to give rise to GFAP+ tumor cells, or 2) GFAP+ cells that express the *Math1-Cre* transgene become transformed by *SmoM2* to give rise to GFAP+ tumor cells.

Our paper puts forward the first of these explanations, which provides a parsimonious interpretation of the data. We considered the second explanation, and we rejected that interpretation based on the vast difference in the fraction of GFAP+ cells that derive from *Math1-Cre* expressing cells in cerebella of mice without *SmoM2* compared to the fraction of GFAP+ cells that derive from *Math1-Cre* expressing cells in *SmoM2*+ tumors.

GFAP+ cells that derive from *Math1-Cre* expressing cells in mice without *SmoM2* (marked by DENDRA2 expression in *Math1-Cre/Pham* mice) are vanishingly rare, hundreds-fold more rare than the proportion in tumors of GFAP+ cells that derive from *Math1-Cre* expressing cells (marked by DENDRA2 expression in *M-Smo/Pham* mice). If all, or even most of GFAP+ tumor cells were the progeny of the vanishingly small population of GFAP+ cells that express *Math1-Cre*, they would need to grow at a much faster rate than the CGNP-derived tumor cells, which are also rapidly proliferating, in order to represent 10% of the tumor at P15. However, we do not find any evidence that GFAP+ tumor cells grow faster over time. Because the GFAP+ DENDRA2+ population is much greater in *M-Smo* tumors than in *Math1-Cre* cerebella, generating 10% of the tumor cells from the small population of non-neoplastic GFAP+ *Math1-Cre* expressing cells would require proliferation sufficiently more rapid as to overtake the population growth of the CGNP-like tumor cells, which is not observed. Additionally, Schuller et al. 2008 (reference 19 in our manuscript) observed that *SmoM2* expression in GFAP+ cells in *GFAP-Cre/SmoM2* mice did not cause widespread glial cell transformation, but rather only produced cerebellar tumors that were found to be medulloblastomas. This prior work argues against a general susceptibility of glial progenitors to transformation by *SmoM2*. Based on these lines of evidence, we consider the most parsimonious explanation for glial cells in our tumors to be descended from *Math1-Cre* expressing predecessors to be that *SmoM2*-driven transformation expands the range of fates within the *Atoh1* lineage to include glial phenotypes.

-- (citing Reviewer 2's prior comment) Making the whole analysis pipeline available to the scientific community would be helpful.

(Our response to this prior comment) We have made the code available at: github.com/ben-babcock/Gershon_single-cell, and we have added this link to the

Supplemental Methods. We have also added a link to a web-based application (gershon-lab.med.unc.edu/single-cell/) that allows readers to plot the expression of any gene across the t-SNEs for the WT P7 cerebellum and combined treated and untreated tumors. The link to this application has been added to the first paragraph of the Discussion and to the Data Availability statement.

The link the authors indicated did not show the pipeline. It's only the visualization segments. They need to explain the details of the whole analysis pipeline.

We appreciate the need to explain all of the methods and to make the methods available to the scientific community. For this reason, we made the entire analysis pipeline available through the provided github link. Our code is fully documented on github, as described in our response in the prior round of review. Following the workflow that we provide on github will completely reproduce our analysis including, but not limited to, visualization segments.

The link to the web-based visualization tool was not offered in place of the requested documentation, but rather as an additional method of promoting transparency by making thousands of patterns of gene expression readily available to readers and reviewers without coding expertise.

We accept that our documentation in github could be improved through additional commentary. In order to increase the accessibility, we have added additional commentary to the github post, which will facilitate implementation.

Reviewer #5 (Remarks to the Author): Recruited to replace Reviewer #3; expertise in single cell sequencing

The manuscript "Single-cell RNA-seq shows cellular heterogeneity and lineage expansion in a mouse model of SHH-driven medulloblastoma support resistance to SHH inhibitor therapy" is a timely and well designed study. The team uses single cell RNA-seq in a novel and creative way and their conclusions will be of great interest to the readers.

We appreciate the Reviewer's finding that our study was well-designed and that our application of sc-RNA-seq is novel and of interest.

However, I would like a few comments addressed before publication:

- For the initial vehicle-treated scRNAseq experiments in Figure 2: How many cells were analyzed total and per mouse? How many passed basic QC and how many failed? Based on what parameters (only >500 genes expressed is mentioned in methods) and where were the thresholds? Basic stats and transparency about these data are a must in single-cell analysis.

We agree that transparency in these criteria is essential. In response to the Reviewer's feedback, we have added all of the requested information. Specifically, we added the % of cells passing QC to the Results section (84% in M-Smo tumors and 70% in P7 WT cerebella), and in the Supplementary Materials and Methods we added the rest of the information or clarified some of the information that was already present. The revised text of the Supplementary Materials and Methods now reads:

"We used filtering to identify which putative cells, identified by barcodes, represent informative cells. Putative cells with fewer than 500 UMIs or genes were considered to have too little information to be useful, and potentially to contain mostly ambient mRNA reads. Putative cells with greater than > 4-5 standard deviations above the median nUMI or nGene were suspected to be doublets, improperly merged barcodes, or sequencing artifacts and were excluded. Putative cells with predominantly mitochondrial transcripts (>4-5 standard deviations above the median level of mitochondrial transcripts) were suspected to be dying cells and also excluded. For each QC step based on standard deviation, a cutoff between 4 and 5 standard deviations was selected in order to sample optimally around the mean, as visualized by violin plot following Macosko et al, 2015 guidelines {Macosko, 2015 #1450}. Based on these considerations, QC criteria were:

- UMIs >500 and < 4-5 standard deviations above the median
- Genes >500 < 4-5 standard deviations above the median
- Mitochondrial transcripts < 4-5 standard deviations above the median

In total 84% of putative cells from M-Smo mice and 70% of putative cells from WT mice met QC criteria and were included in the analysis. From the 10 M-Smo mice, including both vehicle-treated and vismodegib-treated animals, we included a total of 29,234 cells, with 5 replicate mice per condition, with a range 642-8062 cells per animal and a median of 2636 cells. From the 5 P7 WT mice, we included a total of 7090 cells, with a range of 788-2049 cells per animal and a median of 1169 cells."

This revision of the description of cell filtering provides all of the requested information.

- Other than clustering and a rather sparse YFP expression, I am not confident about which cells are tumor cells and which are normal cells. Is there a way to call specific mutations or increase the sensitivity of YFP calling? Would these SHH-Mb have CNVs that could be inferred from the data and used as a way to distinguish normal from tumor cells?

The question of how to distinguish tumor cells from normal cells is complex and cell type specific. The neoplastic nature of the large set of neural progenitor-like cells (termed Nodes A_V - C_V in our paper) is newly demonstrated by Seurat CCA analysis that we added in response to the reviewer's suggestion. In this analysis, in contrast to the kNN analysis, cells of Nodes A_V - C_V do not co-cluster with any normal cell types, consistent with our identification of these cells as tumor cells. For these cells, the YFP signal and the CCA provide two different sources of evidence.

For the YFP+ cells that co-cluster with astrocytes and oligodendrocytes in both CCA and kNN studies, however, our lineage trace is the only way to distinguish tumor from non-tumor. While imputation methods such as MAGIC could increase the sensitivity of YFP calling, the additional calls would be based on co-variate genes rather than on direct evidence of YFP expression, and would thus seem not to be definitive for this purpose.

Regarding mutations and CNV analysis, all cells harbor the oncogenic Smo mutation which thus cannot distinguish tumor from non-tumor. CNV analysis, which was used in Tirosh et al 2016 (reference 9 in our manuscript) is not straightforward in our tumors. The tumors in Tirosh et al were oligodendrogliomas, which were known to harbor 1p/19q deletions. In contrast we do not have evidence of CNVs within the tumor. While inferences of CNV may be made in tumors expected to have CNVs, inferring the presence of CNVs, without a priori knowledge that CNVs are present, would seem again to be speculative but not definitive. Because mutation analysis CNV analysis and imputation all seem problematic, we consider that lineage tracing with YFP and DENDRA2 are the most effective ways to identify cells descended from Math1-Cre-expressing predecessors. Based on lineage tracing we assign the YFP/DENDRA2+ glial cells to tumor origin and we consider that all proliferative cells descended from *Math1-Cre* predecessors are tumor cells.

- For the projection of the MB cells onto the normal cerebellum, please use a second method for validation, e.g Seurat CCA.

At the reviewer's request, we have added a Seurat CCA analysis of WT P7 cerebellar cells and vehicle-treated tumor cells (Fig 3d). As described above, this analysis shows that the glial and stromal cells from the M-Smo tumors co-cluster with the same cell types in the WT cerebellum. However, the neural progenitor-like tumor cells of Nodes A_V-C_V do not co-cluster with any normal cell type. As we discuss in the new revision, stromal and glial cell types co-cluster in both the CCA and the kNN, and our kNN method is validated by the correct matching of these cell types in tumor and WT samples. We further state that the separation of the WT CGNPs from the cells of Nodes A_V-C_V in the CCA demonstrates that these tumor cells are clearly different from WT cells. In contrast, the kNN mapping shows that despite the clear differences, the cells of Nodes A_V-C_V are most similar to CGNPs, compare to the other cell types in the P7 WT cerebellum. As discussed below, we now extend the kNN analysis to include recently published datasets from a broader range of WT cells from different ages. By presenting both the CCA and the kNN, we highlight both the differences and the similarities of WT CGNPs and Nodes A_V-C_V cells, and we discuss the different perspectives in the revised text.

- The authors should also apply the mouse MB data to recently published mouse cerebellum data sets, e.g. Carter RA and al, Curr Biol 2018; Hovestadt V. et al, Nature 2019; Vladoiu MC et al, Nature 2019.

At the reviewer's request, we have added a new analysis applying mouse MB data to recently published WT mouse cerebellum data from Carter et al 2018 and Vladoiu et al, 2019. We used the kNN method for this comparison because it effectively identifies the WT cells of greatest similarity to the tumor cells. In contrast, the CCA method did not co-cluster WT progenitors and progenitor-like tumor cells. We subjected the scRNA-seq data from Carter RA et al 2018 and Vladoiu MC et al 2019 to PCA analysis using our pipeline and generated tSNE projections for each dataset.

To identify cell types in these new WT t-SNE projections, we mapped cell type markers that we used in our prior analyses (Sup Fig 2). The domain of CGNPs/CGNs was mapped using the markers *Atoh1*, *Barhl1* and *Grin2b*. In the dataset from Carter et al 2018, stromal cells and glial cells other than astrocytes were removed by the authors prior to our analysis and as expected, our markers identified only the *Atoh1* lineage (CGNP/CGNs/UBCs), other neurons and astrocytes. We then used the kNN method to project P7 WT cells from our studies into the t-SNEs of WT cells from both prior publications (Fig. 3 f,h). P7 WT CGNPs from our analysis mapped to the *Atoh1* lineage clusters, validating our cell type assignments and further localizing the CGNPs within these complex t-SNE projections.

After these preliminary steps, we mapped cells from the *M-Smo* tumors into the WT t-SNEs using the kNN algorithm. As shown in Fig 3g,i glial and stromal cells from *M-Smo* tumors mapped to the same cell types in the Vladoiu MC et al 2019 WT dataset, and astrocytes and CGNs mapped to the correct cell type in the Carter et al 2018 dataset. In both datasets, cells of Nodes A_V-C_V mapped predominantly to the same clusters as the P7 WT cells and follow the axis of CGNP differentiation. A small number of cells from Node A_V and B_V, however, mapped to more primitive cells predating the emergence of CGNPs. We discuss in the Results section that these data show that *M-Smo* tumors predominantly contain cells resembling CGNPs, but that the tumors also contain cells that resemble more primitive, undifferentiated cell types.

We have also added new studies comparing vismo-treated *M-Smo* tumors to WT cells from Carter et al 2018 and Vladoiu MC et al 2019 (Sup Fig 8). These studies show that vismodegib induces a shift toward more differentiated cell types, in agreement with our data from Fig 6. in which tumors from vehicle-treated and vismo-treated *M-Smo* mice are projected by kNN onto WT P7 cells. We thank the reviewer for encouraging these interesting analyses, which strengthen the work.

- I am not sure the last part about PRC2 has strong enough data to support the conclusion in the setting of low gene sensitivity/low coverage per cell. Please address or rather leave out of the paper.

We have removed the figures suggesting a role for the PRC2 complex in vismodegib resistance, to meet the reviewer concern that the data was inconclusive.

REVIEWER #4 COMMENTS

The authors answered all my concerns properly.

I would recommend it for publication and anticipate further functional validations based on this study in their future work.

REVIEWER #5 COMMENTS

The authors have thoroughly and thoughtfully addressed all my concerns and suggestions. I have no further questions and recommend publication of this highly interesting and innovative manuscript.